# Bacterial and host enzymes modulate the pro-inflammatory response elicited by the peptidoglycan of Lyme disease agent *Borrelia burgdorferi*

Joshua W. McCausland[1,2,3☉], Zachary A. Kloos[4☉], Irnov Irnov[1,3☉], Nicole D. Sonnert[4,5☉], Junhui Zhou[6], Rachel Putnik[6], Elizabeth A. Mueller[1,2,3], Allen C. Steere[7], Noah W. Palm[5], Catherine L. Grimes[6], Christine Jacobs-Wagner [1,2,3,8]*

1 Sarafan ChEM-H Institute, Stanford University, Stanford, California, United States of America,
2 Department of Biology, Stanford University, Stanford, California, United States of America,
3 Howard Hughes Medical Institute, Stanford University, Stanford, California, United States of America, 4 Microbiology Program, Yale University, New Haven, Connecticut, United States of America,
5 Department of Immunology, Yale University School of Medicine, New Haven, Connecticut, United States of America, 6 Department of Chemistry and Biochemistry, University of Delaware, Newark, Delaware, United States of America, 7 Center for Immunology and Inflammatory Diseases, Massachusetts General Hospital and Harvard Medical School, Boston, Massachusetts, United States of America, 8 Department of Microbiology and Immunology, Stanford University School of Medicine, Stanford, California, United States of America

☉ These authors contributed equally to this work.
* jacobs-wagner@stanford.edu

## Abstract

The spirochete *Borrelia burgdorferi* causes Lyme disease. In some patients, an excessive, dysregulated proinflammatory immune response can develop in joints leading to persistent arthritis even after antibiotic therapy. In such patients, persistence of antigenic *B. burgdorferi* peptidoglycan (PG^Bb) fragments within joint tissues may contribute to immunopathogenesis pre- and post-antibiotic treatment. In live *B. burgdorferi* cells, the outer membrane shields the polymeric PG^Bb sacculus from exposure to the immune system. However, unlike most diderm bacteria, *B. burgdorferi* releases PG^Bb turnover products into its environment due to the absence of recycling activity. In this study, we identified the released PG^Bb fragments using a mass spectrometry-based approach. By characterizing the L,D-carboxypeptidase activity of *B. burgdorferi* protein BB0605 (renamed DacA), we found that PG^Bb turnover largely occurs at sites of PG^Bb synthesis. In parallel, we demonstrated that the lytic transglycosylase activity associated with BB0259 (renamed MltS) releases PG^Bb fragments with 1,6-anhydro bond on their *N*-acetylmuramyl residues. Stimulation of human cell lines with various synthetic PG^Bb fragments revealed that 1,6-anhydromuramyl-containing PG^Bb fragments are poor inducers of a NOD2-dependent immune response relative to their hydrated counterparts found in the polymeric PG^Bb isolated from dead bacteria. We also showed that the activity of the

**Data availability statement:** Raw MS data are available in the GlycoPOST repository (PMID 33174597) under accession numbers GPST000537 and GPST000595. Raw microscopy images are available in the BioImage Archive (PMID 35189131) under the accession number S-BIAD1573. All other datasets and code for image and MS data analysis can be downloaded from the Jacobs-Wagner lab Github site (https://github.com/JacobsWagnerLab/published/tree/master/McCausland_et_al_2025).

**Funding:** This research was supported in part by funding from the James D. Jamieson and Family M.D.-Ph.D. Scholarship Fund at Yale University (to Z.A.K), the Medical Scientist Training Grant T32 GM007205 from the National Institute of General Medical Sciences, National Institutes of Health (to Z.A.K), the Stanford Bio-X Interdisciplinary Initiatives Seed Grants Program (IIP) [R10-9] (to C.J.-W.), the National Institutes of Health (R01GM138599 to C.L.G), and the Younger Family (to C.J.-W.). C.J.-W. is an investigator of the Howard Hughes Medical Institute. The funders had no role in study design, data collection and analysis, decision to publish, or preparation of the manuscript.

**Competing interests:** The authors have declared that no competing interests exist.

human *N*-acetylmuramyl-L-alanine amidase PGLYRP2, which reduces the immuno-genicity of PG$^{Bb}$ material, is low in joint (synovial) fluids relative to serum. Altogether, our findings suggest that MltS activity helps *B. burgdorferi* evade PG-based immune detection by NOD2 during growth despite shedding PG$^{Bb}$ fragments and that PG$^{Bb}$-induced immunopathology likely results from host sensing of PG$^{Bb}$ material from dead (lysed) spirochetes. Additionally, our results suggest the possibility that natural variation in PGLYRP2 activity may contribute to differences in susceptibility to PG-induced inflammation across tissues and individuals.

## Author summary

During bacterial infection, the presence of peptidoglycan—a polymeric element of bacterial cell walls—triggers a host inflammatory response. Although generally protective during acute phases, inflammation, when chronic, can contribute to disease development. Recent work has suggested that the persistence of pro-inflammatory peptidoglycan derived from the Lyme disease spirochete *Borrelia burgdorferi* in joints may contribute to persistent arthritis in some patients despite appropriate antibiotic therapy. Interestingly, *B. burgdorferi* sheds peptidoglycan turnover products into the environment during growth. Here, we show that these shed products from live spirochetes are poor effectors of an immune response by the human NOD2 immune receptor due to the formation of an anhydro bond on the *N*-acetyl-muramic residue during peptidoglycan hydrolysis by a *B. burgdorferi* lytic transglycosylase. We also show that human *N*-acetylmuramyl-L-alanine amidase activity, which abrogates the NOD2-dependent response to the immunogenic peptidoglycan isolated from lysed *B. burgdorferi* cells, is low in human joint fluids relative to serum. Based on our findings, we propose that immunopathogenesis by peptidoglycan material more likely derives from lysed spirochetes (killed by an immune attack or antibiotics) than live ones and that the level of human peptidoglycan hydrolytic enzymes across tissues and individuals influences susceptibility to chronic inflammation.

## Introduction

Lyme disease is the most prevalent and fastest-growing vector-borne disease in the United States and Northern Europe [1–4]. It is caused by *Borrelia burgdorferi*, a spirochete that is transmitted by *Ixodes* ticks. This pathogen circulates in nature through transmission and persistence in vertebrate reservoirs including rodents, birds, and lizards [5]. Infection in humans is often apparent by the presence of a slowly expanding skin lesion at the site of the tick bite. In the absence of appropriate diagnosis and antibiotic therapy, *B. burgdorferi* can disseminate widely to extracutaneous tissues, producing multisystem clinical manifestations such as flu-like symptoms, meningitis, carditis, and often arthritis, a late disease manifestation.

*B. burgdorferi* is not known to produce toxins that damage host cells. Rather, clinical symptomatology is caused by the host inflammatory response [6]. In some patients with Lyme arthritis, an exaggerated, dysregulated proinflammatory immune response develops, which can lead to persistent arthritis even after appropriate antibiotic therapy. This response is often referred to as postinfectious, antibiotic-refractory Lyme arthritis. The consequences in joint (synovial) tissues, which are the target of the immune response, include vascular damage, autoimmune and cytotoxic reactions, and massive fibroblast proliferation and fibrosis [7]. What leads to the persistence of these symptoms is not well understood and likely involves multiple factors. We recently proposed peptidoglycan (PG) as one of these potential contributing factors [8]. PG, an essential component of bacterial cell walls and a well-known immunogen [9], is made of glycan strands of repeating disaccharide-peptide units that form a crosslinked polymeric structure (known as a sacculus) around the bacterial cytoplasmic membrane. *B. burgdorferi* PG (PG$^{Bb}$) has been demonstrated to be immunogenic, as purified PG$^{Bb}$ sacculi induce an inflammatory response in mammalian cell cultures, C3H/HeJ mice, and humans [8,10]. Remarkably, evidence suggests that PG$^{Bb}$ material can persist in the synovial fluids of Lyme arthritis patients, even months to a year after antibiotic treatment [8]. Furthermore, these patients have a high level of anti-PG$^{Bb}$ IgG antibodies detected within the synovial fluid of the affected joint [8]. In the host, PG$^{Bb}$ is detected by the innate immune receptor NOD2 [8]. Interestingly, the transcript level of NOD2 transcripts is elevated in the inflamed synovial tissues of Lyme arthritis patients long after the resolution of the acute infection [11]. In addition, mouse and in vitro cell experiments have linked NOD2 to proinflammatory cytokine production and immune tolerance during *B. burgdorferi* infection [12,13]. Altogether, these observations have led to the hypothesis that the persistence of residual PG$^{Bb}$ contributes to the pathogenesis of Lyme arthritis [8] by acting as an adjuvant that stimulates chronic inflammation and/or by bolstering T cell proinflammatory immunoreactivity with autoantigens [14].

In live *B. burgdorferi* cells, the outer membrane shields the PG$^{Bb}$ sacculus from immune detection by the host. However, two potential sources of PG$^{Bb}$ exposure to the immune system have been proposed [8]. First, when *B. burgdorferi* cells die due to the harsh conditions in the host (e.g., innate immunity or antibiotic treatment), they lyse, releasing PG$^{Bb}$ material to the surrounding environment. Second, live spirochetes degrade ~40% of their PG during growth but, unlike *Escherichia coli* and many other diderm bacteria [15], *B. burgdorferi* does not recycle PG turnover products during growth [8]. As a result, growing *B. burgdorferi* cells continuously release PG$^{Bb}$ fragments into the culture supernatant, possibly through diffusion across the outer membrane through porins.

These observations raise a number of questions. Do PG$^{Bb}$ materials released from both dead and live spirochetes contribute to immunopathogenesis? Or is one source more likely to contribute than the other? If live *B. burgdorferi* does indeed release PG$^{Bb}$ fragments into the environment during growth, how does this pathogen evade PG-based immune recognition and eradication by the host? Could the PG$^{Bb}$ material shed from live *B. burgdorferi* cells be chemically distinct from the immunogenic PG$^{Bb}$ of dead cells to avoid detection? Do PG hydrolases produced by *B. burgdorferi* and/or the host play a role in modulating immune response and pathogenesis? We sought to address these questions by characterizing the PG$^{Bb}$ fragments released by live *B. burgdorferi* cells during growth and by identifying both bacterial and host factors that modulate PG$^{Bb}$-induced immune responses.

## Results

### Multiple *B. burgdorferi* strains shed peptidoglycan fragments of the same composition during growth

During growth, new disaccharide-peptides are incorporated into the existing PG sacculus. In *B. burgdorferi*, these PG precursors consist of *N*-acetyl-glucosamine (Glc*N*Ac) and *N*-acetyl-muramic acid (Mur*N*Ac) linked to a peptide chain made of L-alanine (L-Ala), D-glutamic acid (D-Glu), L-ornithine (L-Orn), and two D-alanine residues (D-Ala), with L-Orn being connected to a single glycine (Gly) [8,10]. The growing PG polymer is then crosslinked, remodeled, and trimmed by PG enzymes, leading to the mature sacculus illustrated in Fig 1A [8].

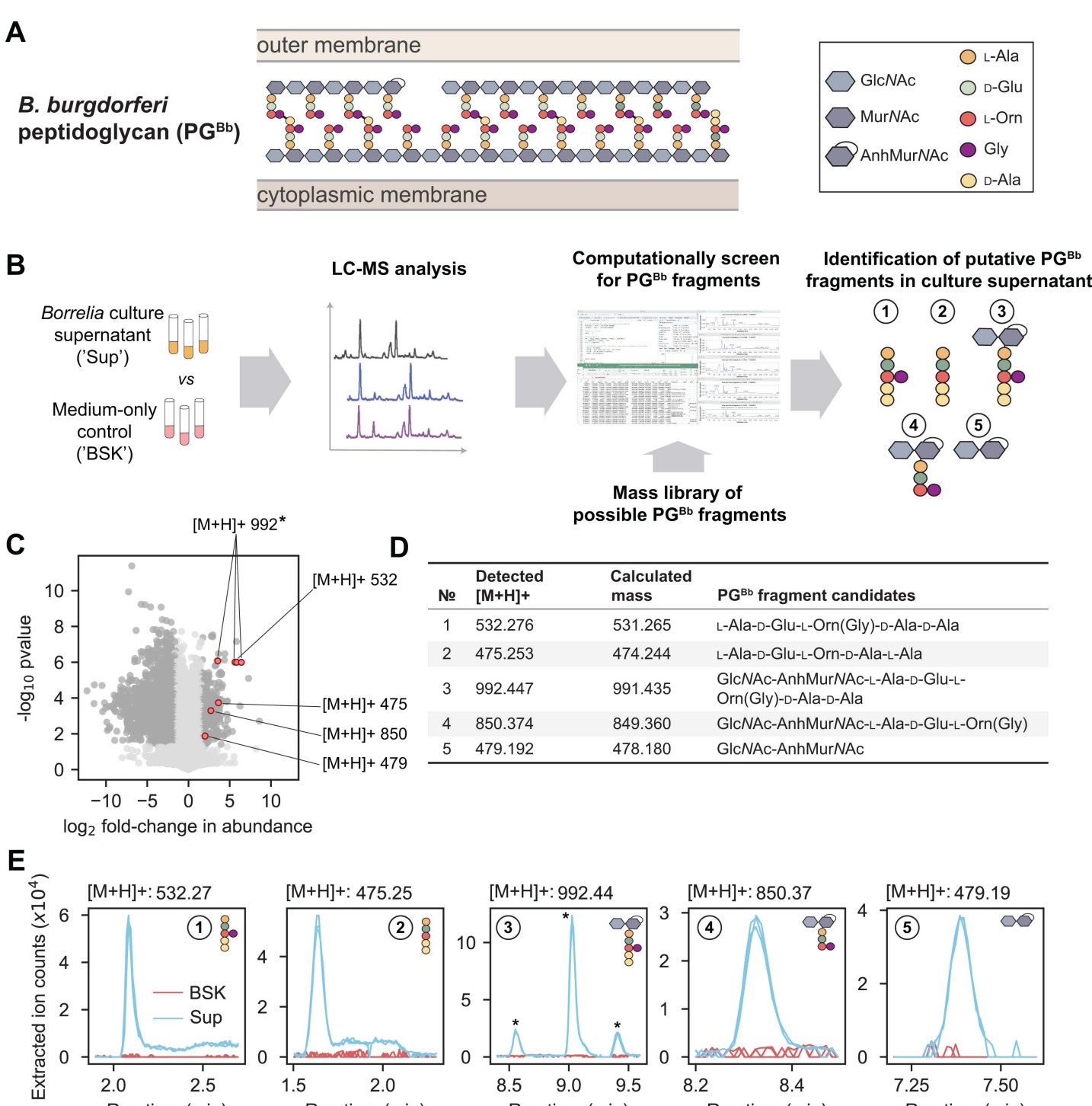

**Fig 1. Mass spectrometry of *B. burgdorferi* culture supernatant identifies five peptidoglycan monomeric species. A.** Schematic showing the chemical composition of the *B. burgdorferi* peptidoglycan (PG^Bb) and its location between the cytoplasmic and outer membranes. **B.** Workflow of the experimental approach: the supernatants ('Sup') of *B. burgdorferi* cultures vs. medium-only controls ('BSK') were analyzed by LC-MS to identify enriched mass features. Any enriched hits were compared against a mass library of potential PG^Bb monomers and dimers from which five candidate PG^Bb fragments were identified. **C.** Volcano plot of [M+H] features either enriched or depleted compared to medium only. Dark gray dots are mass features that are below the false discovery rate threshold of *p<0.05* and are greater than a fold-change of 3. Mass features that correspond to five candidate PG^Bb fragment candidates are indicated in red and labeled with the corresponding [M+H]+ values. The asterisk marks the PG^Bb fragment candidate with

the same [M+H]+ value (992), but with three differently enriched isomers. The datapoint for one of these isomers overlaps with a datapoint for another putative PG$^{Bb}$ fragment ([M+H]+=532). **D.** Table of predicted (calculated from our theoretical PG$^{Bb}$ fragment library, S1 Dataset) and detected masses of the five candidate PG$^{Bb}$ fragments identified in the culture supernatants. **E.** Individual extracted ion chromatograms (EICs) showing the enrichment of the PG$^{Bb}$ fragment species in the supernatant (Sup) relative to medium-only control (BSK). The asterisks show the three peaks corresponding to isomeric forms of PG$^{Bb}$ fragment #3.

For this study, our first goal was to identify the major species of PG$^{Bb}$ fragments that *B. burgdorferi* releases into its culture medium during growth. This was a non-trivial task, as laboratory cultures of *B. burgdorferi* require the use of a highly complex, undefined medium (e.g., BSK-II, which includes GlcNAc, glucose, CMRL medium, bovine serum albumin, rabbit serum, neopeptone, and yeastolate). To find rare PG$^{Bb}$ fragments in a complex mixture of other molecules in BSK-II, we developed a pipeline in which mass features enriched in the culture supernatant over the cell-free growth medium are screened against a library of simulated PG$^{Bb}$ fragment masses (Fig 1B, see Materials and Methods). Briefly, replicates of a clonal derivative of the *B. burgdorferi* type strain B31 (B31 IR) were grown in BSK-II to late exponential phase (i.e., density of ~5 x 10$^7$ cells/mL). Culture supernatants ('Sup') were then harvested and analyzed by liquid chromatography (LC) coupled to mass spectrometry (MS) in comparison to the cell-free medium-only control ('BSK').

We found that over 2000 mass features displayed a significant ($p<0.01$, Welch $t$-test and FDR-corrected) difference abundance (at least three-fold) between culture supernatants and the BSK-II control (Fig 1C). Most of these mass features were underrepresented in the culture supernatants, as expected from nutrients consumed from the BSK-II medium during *B. burgdorferi* growth. Of greater interest were the 270 mass features found to be enriched in the culture supernatants. To examine whether any of them might correspond to PG$^{Bb}$ fragments, we computationally generated a mass library representing more than 700 permutations of possible PG$^{Bb}$ monomers and dimers (S1 Dataset and Fig 1B). Of the compounds that were undetectable in medium-only samples but present in culture supernatants, we found five molecules with masses corresponding to predicted monomeric PG$^{Bb}$ species (Fig 1D and S1 Table, PG fragment species #1–5). One of them ([M+H]+=992) was represented three times in Fig 1C due to the presence of isomers (marked by asterisks in Fig 1E). Two of the identified putative PG$^{Bb}$ fragment candidates were peptide stem variants of the PG$^{Bb}$: one with Gly linked to Orn, L-Ala-D-Glu-L-Orn(Gly)-D-Ala-D-Ala (PG fragment #1), and the other without Gly, L-Ala-D-Glu-L-Orn-D-Ala-D-Ala (PG fragment #2). Two others were disaccharide-peptides: GlcNAc-AnhMurNAc-L-Ala-D-Glu-L-Orn(Gly)-D-Ala-D-Ala (PG fragment #3) and GlcNAc-AnhMurNAc-L-Ala-D-Glu-L-Orn(Gly) (PG fragment #4). The remaining molecule consisted of the disaccharide moiety alone: GlcNAc-AnhMurNAc (PG fragment #5). All three sugar-containing candidates (PG fragments #3–5) were found to harbor MurNAc modified with an intramolecular 1,6-anhydro bond (Anh). This anhydro bond corresponds to the loss of a water molecule (18 Da). Plotting the extracted ion counts of the identified molecules demonstrated that all five PG fragment candidates were in quantities below the level of detection in the BSK II-only control, yet present in each of the three replicates culture supernatant samples analyzed (Fig 1E). The identity of each candidate was confirmed by tandem MS fragmentation (S1–S5 Figs).

We found that these PG$^{Bb}$ monomers accumulated during *B. burgdorferi* growth in BSK-II medium (Fig 2A and 2B), consistent with these molecules being the products of PG$^{Bb}$ metabolism. This was observed not only for the strain B31 IR, our clonal derivative of the type strain B31 MI, but also for other *B. burgdorferi* strains, including K2, a clonal derivative of B31 MI amenable for cloning [16,17], the infectious patient isolate strain 297 [18], and the infectious tick isolate N40 [19]. We verified that B31 IR, K2, and N40 strains all have the expected complement of plasmids originally described for each strain (S6A–S6C Fig) [17,20]. Our strain 297 lacks cp32–6 and cp32–9 (S6D Fig) compared to the plasmid set previously described [20]. We expect this strain to remain infectious as recent work has shown that cp32 plasmids are dispensable for infectivity in mice [21]. A similar growth-dependent collection of PG$^{Bb}$ fragments was found to accumulate in the culture medium of all four *B. burgdorferi* strains tested (Fig 2B).

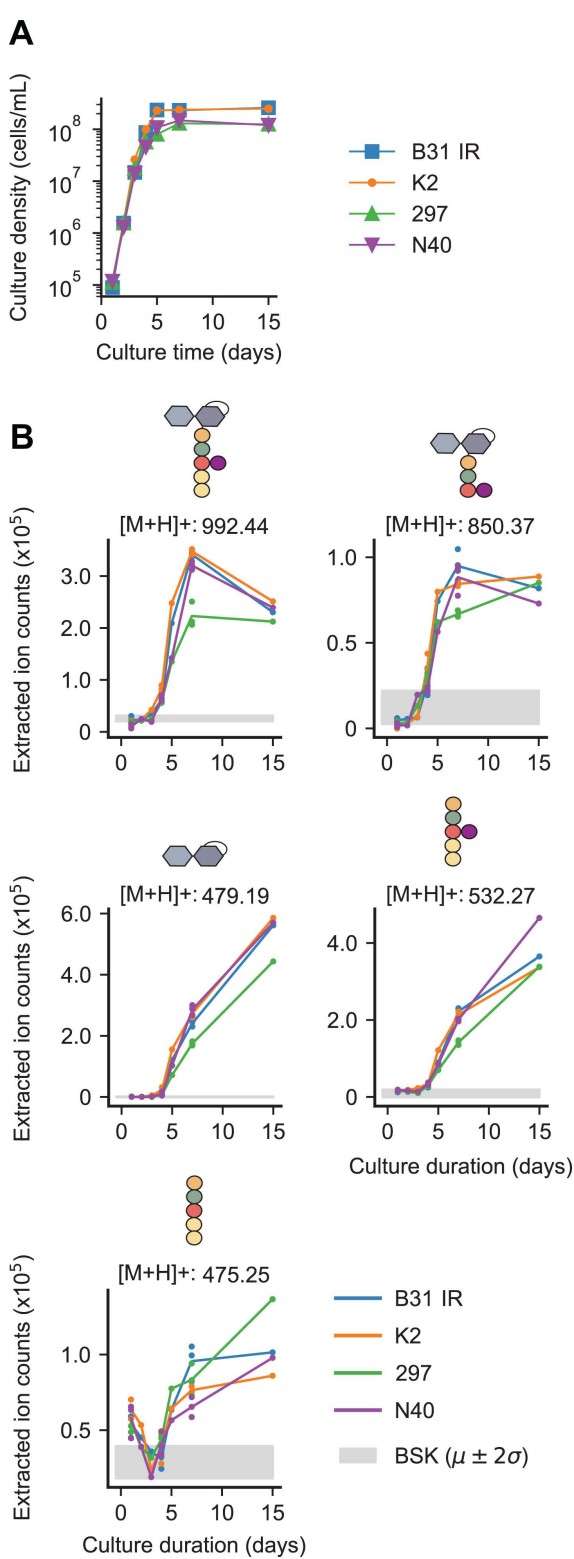

**Fig 2. Various *B. burgdorferi* strains accumulate PG^Bb monomers in their culture supernatants during growth. A.** Growth curves of four indicated *B. burgdorferi* strains. **B.** Plots showing the accumulation (increase in extracted ion count) of the indicated PG^Bb fragment species in culture supernatants over time. Strains B31 IR, K2, 297, and N40 were sampled once daily during normal growth starting at a density of 1x10^4 cells/mL. Three biological

replicates are included for days 1, 4, and 7, while a single measurement was taken for days 2, 3, 5, and 15. The Grey shade shows the mean (μ) ± two standard deviations (2σ) for the background detection in the medium-only control (BSK).

## BSK-II medium and one of its components, heat-inactivated rabbit serum, can possess residual peptidoglycan-hydrolytic activity

We noted that the levels of peptide- and disaccharide-only PG$^{Bb}$ fragments continued to accumulate after the cultures reached stationary phase around day 7 (Fig 2B). In contrast, the levels of disaccharide-peptides often leveled off or decreased in these stationary phase cultures (Fig 2B). While these observations may reflect changes in PG enzymatic activities in stationary-phase *B. burgdorferi* cells, it also raised the possibility that the BSK-II medium itself may possess *N*-acetylmuramyl-L-alanine amidase activity that results in the cleavage of some shed disaccharide-peptides into their constituent parts (i.e., disaccharide- and peptide-only fragments). BSK-II growth medium contains rabbit serum at a final concentration of 6% (v/v), with standard heat inactivation of complement activity while preserving growth-promoting factors. Since rabbit serum has been reported to contain *N*-acetylmuramyl-L-alanine amidase activity [22], we reasoned that our heat-inactivated rabbit serum may still possess trace of amidase activity. If this were the case, incubation of purified PG$^{Bb}$ sacculi in complete BSK-II medium or in 100% heat-inactivated rabbit serum alone would be expected to result in the liberation of peptide-only PG fragments. Indeed, as shown by LC-MS analysis (S7 Fig), we detected the accumulation of L-Ala-D-Glu-L-Orn(Gly) peptides, the prevalent peptide stem in the mature PG$^{Bb}$ sacculus (Fig 1A) [8], in both BSK-II and heat-inactivated rabbit serum. These results suggest that at least a fraction of the disaccharide and peptide-only PG$^{Bb}$ fragments found in the culture supernatant is generated by mammalian enzymes present in the BSK-II medium.

## Peptidoglycan synthesis and turnover are largely coupled during *B. burgdorferi* growth

The prevalence of the peptide stem carrying a terminal D-Ala-D-Ala dimer among the PG$^{Bb}$ fragments present in culture supernatants was surprising. D-Ala-D-Ala is predicted to be rare in the mature PG$^{Bb}$ layer (Fig 1A) as it is not even detected in the chromatogram of muramidase-digested PG$^{Bb}$ sacculi isolated from cells grown in vitro [8]. Instead, unlinked peptide stems in the mature PG$^{Bb}$ are almost exclusively of the form L-Ala-D-Glu-L-Orn(Gly) (Fig 1A). This suggests that the larger L-Ala-D-Glu-L-Orn(Gly)-D-Ala-D-Ala peptide stems present in PG$^{Bb}$ precursor molecules are rapidly either crosslinked or trimmed by carboxypeptidases (CPases) to remove the terminal D-Ala residues and regulate the degree of crosslinking between glycan strands [23,24]. Accordingly, the PG$^{Bb}$ sacculus of a *B. burgdorferi* mutant defective in CPase activity would be expected to contain a higher proportion of peptide stems with terminal D-Ala residues.

We obtained a *B. burgdorferi* mutant (strain T08TC493) with a Himar*1* transposon insertion at position 300 downstream of the first codon of the *bb0605* open reading frame, which is predicted to encode a CPase of 406 amino acids [25]. We purified PG$^{Bb}$ sacculi from *bb0605*::Himar*1* cells and compared their chemical composition to that of sacculi isolated from the B31-derived parent strain 5A18NP1 following enzymatic digestion of both sacculus preparations using mutanolysin, a muramidase that cleaves the β(1–4) glycosidic bond between MurNAc and GlcNAc. The total-ion-count profiles representing digested sacculi from *bb0605*::Himar*1* and its parent differed considerably (Fig 3A). Note that each PG$^{Bb}$ fragment typically eluted at more than one retention time, reflecting different isomeric forms. The extracted ion count profiles obtained for the parent strain revealed PG monomers primarily harboring peptide stems lacking D-Ala residues (variants i and ii in Fig 3B), consistent with the results of a previous PG$^{Bb}$ digest analysis performed on another *B. burgdorferi* strain [8]. In contrast, *bb0605*::Himar*1* sacculi demonstrated decreased abundance of such peptide stem variants (Fig 3B); instead, they were associated with a high proportion of PG monomers carrying longer peptide stems that contained D-Ala-D-Ala (variants iii and iv in Fig 3C). These data strongly suggest that BB0605 is an L,D-carboxypeptidase that prunes terminal D-Ala residues from PG$^{Bb}$ peptide stems in wild-type cells. We propose this protein to be renamed D-Ala carboxypeptidase A (DacA) based on structural prediction comparison with an *E. coli* homolog of the same name (S8 Fig) [26].

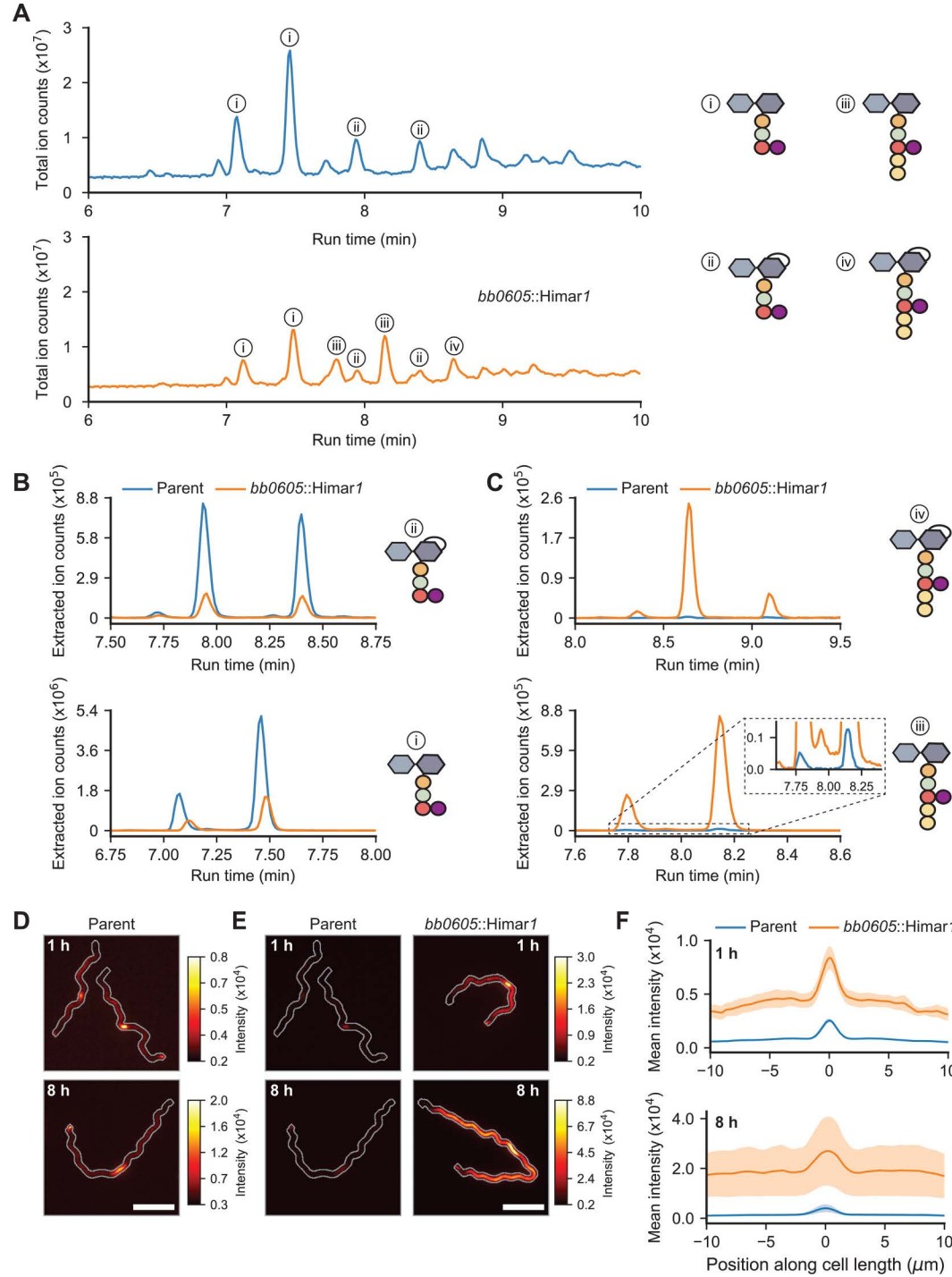

**Fig 3. BB0605 encodes an ʟ,ᴅ-carboxypeptidase. A.** Total ion chromatograms of parent (strain 5A18NP1) vs. *bb0605*::Himar*1* (strain T08TC493) sacculi digested with mutanolysin. Relevant peaks are denoted with numbers corresponding to the most abundant PG[Bb] fragment species. **B.** Plots showing the extracted ion count (EIC) of GlcNAc-AnhMurNAc-ʟ-Ala-ᴅ-Glu-ʟ-Orn(Gly) (top) and GlcNAc-MurNAc-ʟ-Ala-ᴅ-Glu-ʟ-Orn(Gly) (bottom) for the parent vs. *bb0605*::Himar*1* strains as a function of their run time through the liquid chromatography column. **C.** Same as (B) but for GlcNAc-AnhMurNAc-ʟ-Ala-ᴅ-Glu-ʟ-Orn(Gly)-ᴅ-Ala-ᴅ-Ala (top) and GlcNAc-MurNAc-ʟ-Ala-ᴅ-Glu-ʟ-Orn(Gly)-ᴅ-Ala-ᴅ-Ala (bottom). **D.** Representative fluorescent images of parent cells stained with HADA for 1 and 8 h. Cell outlines were generated by making a mask of the cell in the corresponding phase contrast image, dilating it five times, and creating a contour of the binary mask. Scale bar is 5 μm. **E.** Same as (D) except that *bb0605*::Himar*1* cells are shown and that the HADA

signal for parent cells are contrast-normalized to that of *bb0605*::Himar*1* cells. The look up table on the right applies to images with the same time point. **F.** Mean linescan intensities of HADA staining in parent vs. *bb0605*::Himar*1* cells. The shaded area represents the standard error of the mean (n = three biological replicates for each time point for which over 200 cells were analyzed per condition).

The LC-MS experiments with the *bb0605*::Himar*1* mutant also allowed us to determine the retention time of PG$^{Bb}$ monomers harboring terminal D-Ala-D-Ala and verify that sacculi of the parent strain do indeed contain a small amount of these monomeric subunits (inset, Fig 3C). This small amount of D-Ala-D-Ala-containing peptide stems likely correspond to PG$^{Bb}$ precursor molecules newly incorporated into the existing sacculus (i.e., before crosslinking and processing). This is consistent with our previous work showing that zones of new PG synthesis in *B. burgdorferi* are marked by fluorescent D-Ala analogs such as 7-hydroxycoumarincarbonylamino-D-alanine (HADA) upon their substitution for native D-Ala at the fifth amino acid position of the peptide stem [27]. We reasoned that if the HADA signal previously observed in wild-type *B. burgdorferi* cells represents the low level of newly inserted D-Ala-D-Ala-containing peptide stems in the PG$^{Bb}$ layer, this signal should increase considerably in the *bb0605*::Himar*1* mutant cells. As we reported previously [27], the addition of HADA to parent cell cultures for 1 h resulted in preferential accumulation of HADA signal at midcell due to enriched PG$^{Bb}$ growth at that site, with occasional polar enrichment derived from division (Fig 3D). We observed the same overall spatial pattern of HADA labeling in *bb0605*::Himar*1* cells, but at a greater level, as shown qualitatively in images (Fig 3E) and by fluorescence signal quantification along the cell length (Fig 3F). This is consistent with an increased level of D-Ala-D-Ala-containing peptide stems in the PG$^{Bb}$ layer in the absence of DacA activity. Increasing the time of HADA incorporation from 1 to 8 h (the latter approximating the generation time of *B. burgdorferi*) further increased the fluorescence signal intensity into the PG$^{Bb}$ due to growth (Fig 3D–3F). Altogether, our data suggest that, in a wild-type context, a considerable fraction of PG$^{Bb}$ fragments shed into the environment is derived from D-Ala-D-Ala-containing PG$^{Bb}$ material that had just been incorporated, before the D-Ala are removed by DacA or cross-linking. This indicates that the degradation of the PG is tightly coupled in time and space with its synthesis in *B. burgdorferi*.

## BB0259 (MltS) contributes to the shedding of AnhMur*N*Ac-containing peptidoglycan monomers through lytic transglycosylase activity

Another characteristic of the released PG fragments we identified was that all three sugar-containing molecules contained AnhMur*N*Ac. No corresponding molecules containing Mur*N*Ac residues in hydrated/reducing form (i.e., no anhydro bond) were detected in culture supernatants (S9 Fig). AnhMur*N*Ac-containing PG fragments are the product of bacterial enzymes known as lytic transglycosylases (LTGases) [28]. These enzymes cleave the β(1,4)-glycosidic linkage between Glc*N*Ac and Mur*N*Ac residues and, in the process, catalyze an intramolecular transglycosylation reaction that leads to the formation of an anhydro bond between $C_1$ and $C_6$ of Mur*N*Ac [28–31]. When these enzymes act on the polymeric PG, the anhydro bonds they introduced into the PG sacculus prevent the incorporation of another disaccharide-peptide, thereby controlling the average length of glycan strands within the sacculus [28,29,32–34]. Thus, AnhMur*N*ac residues are found only at the end of glycan strands (Fig 1A) and are, as a consequence, found in low abundance in the PG layer relative to Mur*N*Ac residues. The activity of LTGases is also important for degrading large PG turnover products into small soluble fragments [35].

Based on sequence homology searches, *B. burgdorferi* has two putative LTGases [36]. One of them, encoded by the *bb0259* gene, is required for the motility and wavy morphology of *B. burgdorferi* cells [36]. This putative LTGase has been proposed to degrade PG$^{Bb}$ above the site of flagellar basal body insertion into the cytoplasmic membrane to allow flagellar filament assembly across the PG$^{Bb}$ layer [36]. Based on gene deletion results, the other putative LTGase-encoding gene, *bb0531,* has no apparent effect on flagellar assembly, motility, or cell morphology [36]. Since the LTGase activity had not previously been demonstrated for either gene product, we obtained *bb0259* and *bb0531* knockout strains [36] and examined their ability to release AnhMur*N*Ac-containing PG fragments into culture media relative to the parental strain. LC-MS

revealed that the levels of all three AnhMur*N*Ac-containing PG turnover products present in the culture supernatants of the parent strain were considerably lower in the supernatants of the Δ*bb0259* cultures, but not in those of the Δ*bb0531* cultures (Fig 4). Furthermore, the levels of released AnhMur*N*Ac-containing molecules were largely restored in the Δ*bb0259* strain expressing wild-type *bb0259* from a shuttle vector (Fig 4), indicating phenotypic complementation. These results provide compelling evidence that *bb0259* encodes a bona fide LTGase that contributes to PG turnover and shedding in *B. burgdorferi*. Based on structural prediction, this *B. burgdorferi* LTGase shares similarity with the cytosolic *E. coli* Stl70 (S10 Fig). However, BB0259 differs from Stl70 by being anchored to the inner membrane [37]. Therefore, in consultation with Dr. Motaleb whose laboratory at East Carolina University generated the Δ*bb0259* strain used in this study [36], we propose the *bb0259* gene to be renamed *mltS* and its product, MltS, for "membrane-bound lytic transglycosylase in spirochetes."

### The peptidoglycan monomers released during *B. burgdorferi* growth generate little to no NOD2-dependent inflammatory response in vitro

Our findings indicate that the PG^Bb fragments shed by live spirochetes during growth display chemical differences relative to the polymeric PG^Bb sacculus, which can become exposed when *B. burgdorferi* dies and lyses. How these chemical variations may affect the host immune response is not entirely clear. For instance, purified anhydromuramyl derivatives of Mur*N*Ac-peptides have been shown to be poor inducers of NOD2 [38,39]. On the other hand, the AnhMur*N*Ac-containing PG fragment GlcNAc-AnhMur*N*Ac-L-Ala-D-Glu shed by *Neisseria* species has been described as a NOD2 agonist [40].

To examine this in the context of *B. burgdorferi* pathogenesis, we used a panel of synthetic PG^Bb fragments (30 µM) with varying chemical features [41]. For simplicity, we will hereafter refer to PG fragments with both AnhMur*N*Ac and

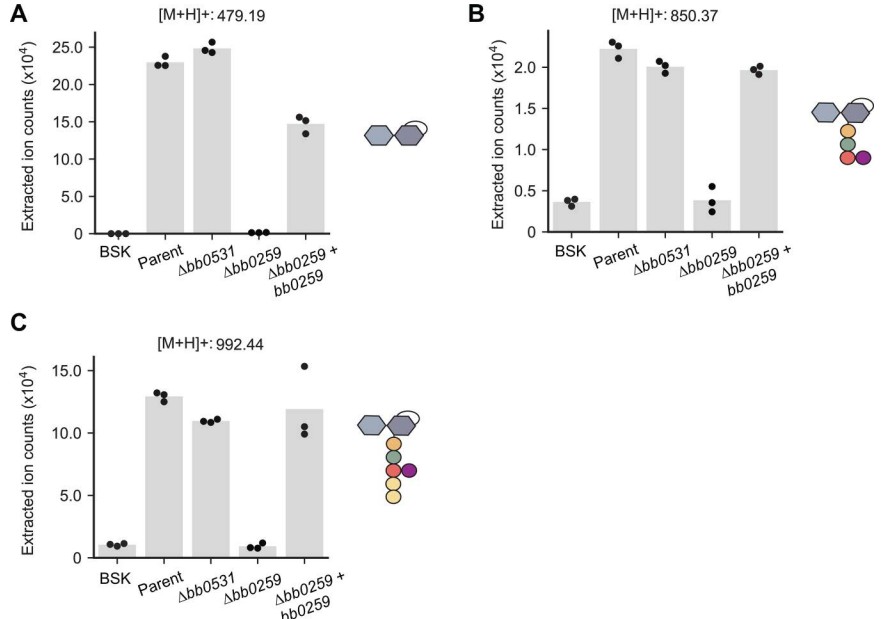

**Fig 4. The lytic transglycosylase BB0259 (renamed MltS) is responsible for the accumulation of AnhMur*N*Ac-containing PG^Bb species in culture supernatants.** For all panels, the medium-only control (BSK) was compared to the culture supernatants of the following strains: parent (B31-A), Δ*bb0531*, Δ*bb0259* and Δ*bb0259* complemented by expression of *bb0259* from the *flgB* promoter on a shuttle vector [36]. **A.** Plot showing the extracted ion counts (EIC) for GlcNAc-AnhMur*N*Ac in the culture supernatants of the indicated strain. Dots represent data from three biological replicates, and the height of each bar represents the mean. Each EIC value was obtained by integrating the relevant peak for each PG^Bb fragment species. **B.** Same as (A) but for GlcNAc-AnhMur*N*Ac-L-Ala-D-Glu-L-Orn(Gly) **C.** Same as (A) but for GlcNAc-AnhMur*N*Ac-L-Ala-D-Glu-L-Orn(Gly)-D-Ala-D-Ala.

peptide moieties as anhydromuropeptides to distinguish them from the muropeptides that have reducing MurNAc residues. Our panel of synthetic molecules also included the well-known NOD2 agonist muramyl-dipeptide MDP (MurNAc-L-Ala-D-IsoGln) and its inactive isomer MDP-LL which serve as positive and negative controls, respectively [42,43]. For human cell immunostimulation, we first used human macrophage-like THP-1 cells differentiated using phorbol 12-myristate 13-acetate (PMA). These cells produce elevated levels of the cytokine interleukin-8 (IL-8) when stimulated with MDP [44], which we confirmed (Fig 5A). As expected, the negative control MDP-LL, had no stimulatory effect. All muropeptides (i.e., no anhydro bond) stimulated IL-8 production (Fig 5A). The presence of GlcNAc to form the disaccharide-peptide species did not stimulate IL-8 production to the same extent (Fig 5A). Importantly, all the corresponding anhydromuropeptides displayed little to no stimulatory activity of IL-8 production relative to their corresponding muropeptides ($p = 0.005$, S11A Fig). Peptide-only species also had no detectable stimulatory activity (Fig 5A). In a repeated comparative analysis, we included GlcNAc-AnhMurNAc-L-Ala-D-Glu-L-Orn(Gly)-D-Ala-D-Ala, the shed anhydromuropeptide with the two terminal D-Ala (PG$^{Bb}$ fragment #3 in Fig 1 and S1 Table). This additional analysis confirmed that the presence of an anhydro bond considerably reduced the stimulation of IL-8 production irrespective of the presence or absence of the D-Ala- D-Ala moiety (S11B and S11C Fig).

To examine these effects in the context of NOD2, we tested the ability of key synthetic PG$^{Bb}$ fragments to stimulate NF-κB in the HEK-Blue human NOD2 (hNOD2) reporter cell line relative to the NOD2-null parental cell line from which it was derived. For this experiment, we used two concentrations (1 and 30 µM) of synthetic PG$^{Bb}$ fragments. The positive control (MDP) displayed strong NF-κB activation at both concentrations (Fig 5B), with a saturating absorbance (A$_{655}$) near 1.25 (S11D Fig). As expected, none of the peptides alone resulted in NF-κB activation whereas all tested muropeptides resulted in saturating NF-κB activation, even at the lowest concentration (Fig 5B). Conversely, exposure to the corresponding anhydromuropeptides resulted in little to no effect (Fig 5B). In all cases, the lowest concentration (1 µM) of a muropeptide generated a stronger response than the highest concentration (30 µM) of the corresponding anhydromuropeptide, indicating that the anhydro bond formation on the MurNAc residue created by MltS decreases NOD2-dependent stimulation by at least 30-fold ($p = 0.04$, S12A Fig). Our results imply that through anhydro bond formation, the LTGase activity of MltS helps live *B. burgdorferi* evade NOD2-dependent detection despite continuously shedding PG$^{Bb}$ material during growth.

The low immunogenicity of the shed PG$^{Bb}$ monomers questioned the origin of the NOD2-stimulating NF-κB activity that our laboratory previously found in the supernatant of *B. burgdorferi* cultures [8]. Therefore, we re-examined this activity on hNOD2 reporter cells and found no measurable stimulatory activity from culture supernatants of various *B. burgdorferi* strains compared to the positive control MDP (S12B Fig). The reason for the discrepancy is not clear. It is possible that the previously reported stimulatory activity originated from a gradual leakage of PG$^{Bb}$ material from the sacculi of lysed or membrane-fragilized spirochetes into the culture over time. Leakage of PG material from dead cells may also explain residual NOD2-stimulatory activity previously reported for culture supernatants of *E. coli* and other bacterial species [45]. We expect that PG contamination from dead cells may be more variable for *B. burgdorferi* cultures given that their "health" can easily fluctuate with small variations in the individual components of the complex growth medium, which can differ between batches [46–48]. Furthermore, one of the essential components of the *B. burgdorferi* growth medium is serum, which contains innate immunity factors (e.g., complement system). If incompletely heat-inactivated, these factors may contribute to cell killing or membrane destabilization, which, in turn, could result in the release of immunogenic PG$^{Bb}$ fragments into the culture. However, we cannot rule out the possibility that live *B. burgdorferi* cells do release immunostimulatory PG$^{Bb}$ material below the detection level of our current MS and hNOD2 reporter assays. Regardless, such PG$^{Bb}$ material would likely be a minor released product compared to the poorly immunogenic PG$^{Bb}$ turnover products identified in this study. In contrast, the polymeric PG$^{Bb}$ sacculus isolated from spirochetes after cell lysis is rich in the pro-inflammatory molecular motif MurNAc-L-Ala-D-Glu readily sensed by NOD2 relative to the corresponding anhydro-N-acetylmuramyl form [8]. We found that pro-inflammatory MurNAc-peptide motifs remain abundant in PG$^{Bb}$ sacculi isolated

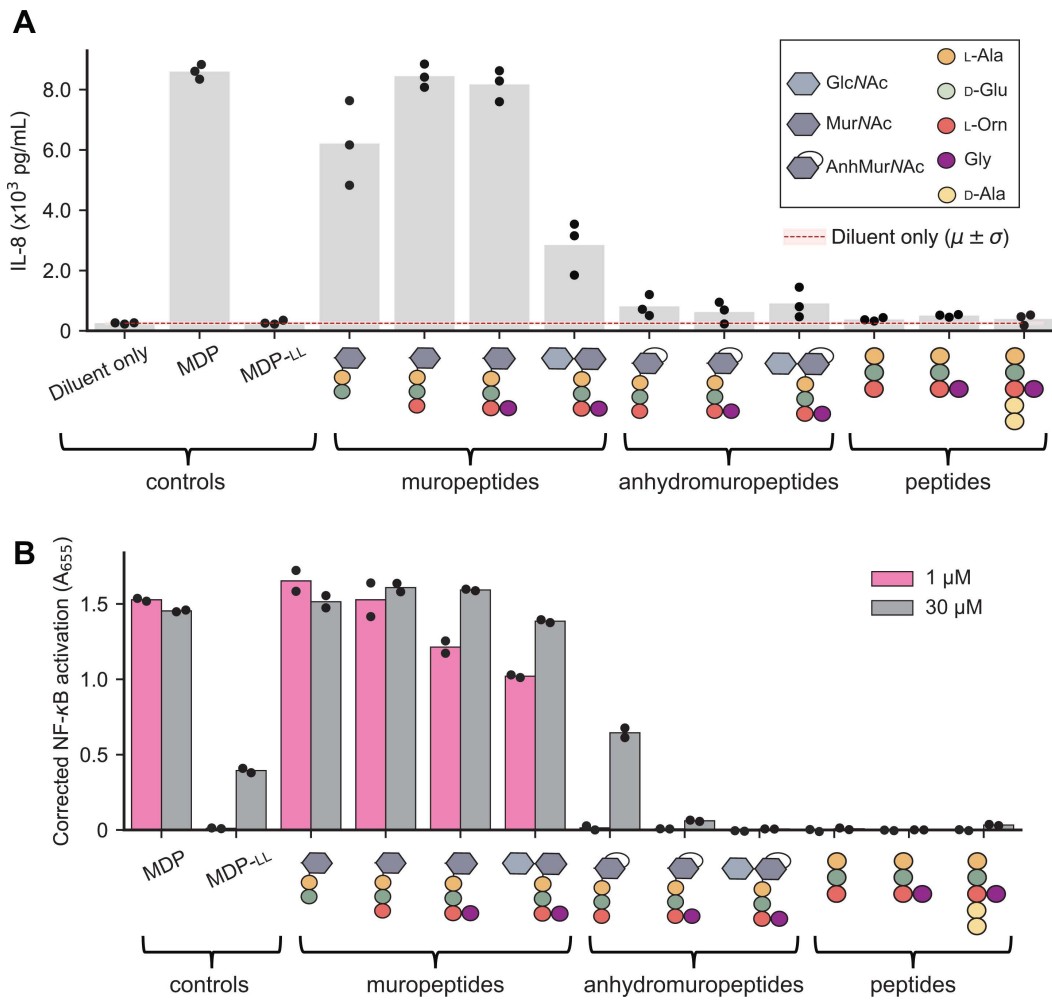

**Fig 5. The *B. burgdorferi* peptidoglycan-induced pro-inflammatory response is abrogated by the presence of AnhMur*N*Ac or the absence of a Mur*N*Ac residue. A.** Plot showing IL-8 production in differentiated THP-1 cells in the presence of different synthetic PG fragments. Bar height represents the mean. Dots represent data from three biological replicates. The legend in the top right defines the schematics for the synthetic PG fragments. A statistical comparison between the muropeptides and the anhydromuropeptides is presented in S11A Fig. **B.** Plot showing secreted embryonic alkaline phosphatase (SEAP) activity, a proxy for NF-κB activation, measured as absorbance at 655 nm ($A_{655}$), in human NOD2 reporter cells (after correction using NOD2-null cells) following stimulation with the indicated molecules. Each dot represents a different biological replicate, and the bar height is the mean of the two replicates. A statistical comparison between the muropeptides and the anhydromuropeptides is presented in S12A Fig.

from 10-day-old stationary phase cultures (S13 Fig) in which >90% of the cells are dead (i.e., have lost their ability to proliferate upon nutrient repletion) [49]. These results suggest that at least under these experimental conditions, the composition of the PG^Bb sacculus remains pro-inflammatory when *B. burgdorferi* dies.

### The levels of host peptidoglycan-hydrolytic activities vary between sera and synovial fluids across Lyme arthritis patients

The immunological results suggest that the primary immunostimulatory forms of PG^Bb derive from the sacculi, presumably when it becomes exposed after cell lysis. Exposed PG^Bb sacculi may then be digested by human hydrolytic enzymes. For instance, lysozyme has muramidase activity that cleaves the β(1,4)-glycosidic bond between Mur*N*Ac and Glc*N*Ac.

But unlike MltS and other bacterial LTGases, lysozyme does not produce an anhydro bond on the MurNAc residues and instead releases muropeptides that are pro-inflammatory (as illustrated in Figs 5 and S11 and S12). Humans can also produce an N-acetylmuramyl-L-alanine amidase, known as PGLYRP2 [50], which can abrogate PGBb-derived inflammatory activity by cleaving between MurNAc and L-Ala to release non-immunogenic peptides. Interestingly, human PG degrading activities can vary considerably across tissues and body fluids [51].

We reasoned that the levels of host PG hydrolytic activity could alter a PGBb-induced immune response depending on the anatomic sites that *B. burgdorferi* colonizes in patients. To examine this idea in the context of Lyme arthritis, we analyzed the digestion products of whole PGBb sacculi incubated for 0 or 6 h in either joint (synovial) fluid samples (n = 6) or serum (n = 4) from Lyme arthritis patients (color-coded in Fig 6) using LC-MS. The patients had a range of inflammatory responses and post-treatment durations of arthritis (see Materials and Methods for details). As an additional control, we analyzed the same set of 0 h incubation samples but without PGBb sacculi added. LC-MS analysis of the 6 h incubation products revealed predicted mass features of PG fragments in the bodily fluids that were virtually absent in the 0 h incubation samples or in the samples lacking PGBb sacculi (Figs 6A and S14A). This indicated that both fluids contained PG-hydrolytic activities. The release of disaccharides alone or in complex with a peptide indicated the presence of lysozyme activity in both fluids. However, the pattern of released fragments was distinct between sera and joint fluids. Incubation in joint fluids produced largely saccharide-peptides and virtually no peptide- or saccharide-only fragments (Fig 6A). In contrast, incubation of *B. burgdorferi* PG sacculi in sera released not only various saccharide-peptide conjugates but also free saccharides and peptides (Fig 6A). The different enrichments of peptide-only masses in sera versus saccharide-peptide monomers in joint fluids are illustrated with a subset of PG fragments in Fig 6B (the complete set is shown in Fig 6A). The results indicate that while both types of fluid display muramidase activity (based on the presence of disaccharides alone or in complex with peptides), they considerably differ in their N-acetylmuramyl-L-alanine amidase activity. Serum samples had significant N-acetylmuramyl-L-alanine amidase activity that cleaved saccharide-peptides into individual saccharides and peptides, whereas joint fluid samples exhibited little, if any, such activity (Figs 6A and 6B and S14B). This was also true when comparing samples from the same (color-coded) patients. We also noted a wide variability in PG-hydrolytic activities across patients (Figs 6A and 6B and S14B). The tissue-specific trends in PG-hydrolytic activities did not appear to be unique to Lyme arthritis patients as they could also be observed in the joint fluids of rheumatoid arthritis patients (n = 3) and the pooled serum of healthy individuals (Figs 6 and S14B).

To examine whether the observed differences in enzymatic activities could at least in part be attributed to differences in levels of lysozyme and PGLYRP2, we measured the abundance of these proteins in the same samples using an enzyme-linked immunosorbent assay (ELISA). This assay showed that the levels of lysozyme and PGLYRP2 in sera or joint fluids were indeed variable across individuals (Fig 6C). Samples with higher muramidase or N-acetylmuramyl-L-alanine amidase activity based on the overall PGBb digest profile (Fig 6A) had a higher concentration in lysozyme or PGLYRP2, respectively (Fig 6C). However, while the trends were preserved, the comparison was not fully consistent, suggesting that other unidentified PG-hydrolytic enzymes may be present in the samples. Importantly, the ELISA confirmed that joint fluids were associated with a generally lower level of PGLYRP2 compared to sera (Fig 6). Collectively, these findings suggest a potential role for human N-acetylmuramyl-L-alanine amidase activity in general and PGLYRP2 in particular in tissue-specific immunopathogenesis induced by lingering PG (as discussed below).

## Discussion

Our study identifies the predominant PG fragments present in the supernatant of *B. burgdorferi* cultures (Fig 1). Our immunological assays (Figs 5 and S11 and and S12) suggest that these PGBb turnover products have low pro-inflammatory activity because of two key chemical features: (1) their peptide stems include an Orn residue at the third amino acid position [8,10] in place of the more common DAP residue found in diderm bacteria [52], and (2) their MurNAc residues carry an anhydro bond linking $C_1$ and $C_6$. The presence of Orn instead of DAP precludes immune recognition by

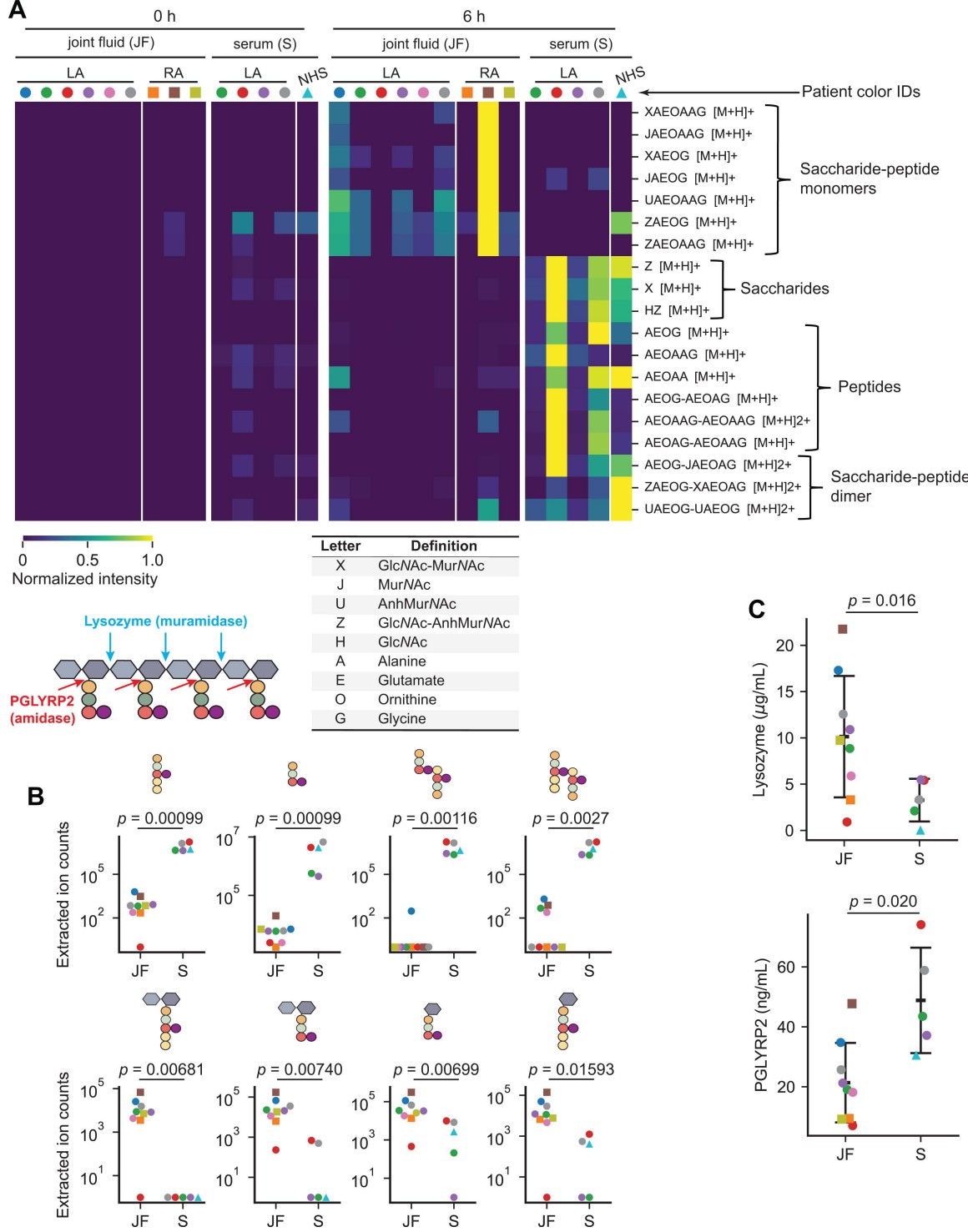

**Fig 6. Incubation of *B. burgdorferi* sacculi in joint fluids and sera of Lyme arthritis patients causes the release of predicted PG<sup>Bb</sup> fragments that vary in composition and amount.** For all panels, the identities (IDs) of Lyme arthritis patients are color-coded. The same color indicates that the serum and joint fluid were obtained from the same individuals. **A.** Heatmap of predicted PG$^{Bb}$ fragment species released from digestion of purified PG$^{Bb}$ sacculi in six joint fluid samples and four serum samples from Lyme arthritis patients. For all samples, sacculi were incubated with either fluid for 0 h (left) or 6 h (right), then the resulting reaction products were analyzed by LC-MS. PG$^{Bb}$ fragment species were detected by their predicted [M+H] value (S1 Table and S1 Dataset). The accompanying table on the left contains the key to interpret the predicted identity of PG$^{Bb}$ fragment species in the heatmap.

Also shown is a schematic illustrating the cut sites for the indicated enzymes. **B.** Representative quantifications of peptide or sugar-peptide conjugate digestion products in serum and joint fluid samples predicted based on their masses. Each dot was derived from integrating the relevant EIC peak. Mann-Whitney U test was used to determine the $p$ values as the data spanned log distances. **C.** ELISA results for either lysozyme or PGLYRP2 for the same joint fluid and serum samples as in (A) and (B). Error bars represent mean ± standard deviation. $p$ values were obtained using a Welch $t$-test since the groups had unequal variances and N values.

the PG-specific receptor NOD1 [53,54] whereas the anhydro bond prevents detection by NOD2. Results from a recent study suggest a molecular mechanism for the latter effect, showing that NOD2 is preferentially activated by Mur$N$Ac-dipeptides that are phosphorylated by the mammalian $N$-acetylglucosamine kinase NAGK [55]. Phosphorylation requires the availability of a hydroxyl group on the $C_6$ position of Mur$N$Ac. The implication, which was not mentioned in the NAGK study [55], is that all anhydromuropeptides produced by bacterial LTGases (including MltS) cannot be phosphorylated by NAGK, as the process of anhydro bond formation eliminates the free hydroxyl group on $C_6$ [31]. Thus, in addition to limiting the length of glycan strands in the PG sacculus [28,29,32–34], LTGase activity also provides an effective means for bacteria to remodel their sacculi during growth without generating PG turnover products detectable by the NOD2 pathway. This conclusion is consistent with previous reports of the immunomodulatory effects of *Helicobacter pylori* and *Neisseria gonorrhoeae* LTGases [38,39]. Given the wide distribution of genes encoding LTGases across diverse bacterial phyla [31], the 1,6-anhydro linkages that these enzymes create in PG turnover products may represent an ancient strategy for evasion of NOD2-dependent immune response. This could particularly be a beneficial strategy for Gram-positive bacteria that most often produce Lys-type PG [52], which, like Orn-type PG, do not trigger a NOD1-dependent immune response [53,54].

Whether the shed PG^Bb turnover products identified herein mediate beneficial interactions between *B. burgdorferi* and its various hosts in nature remains to be determined. Interestingly, anhydromuropeptides have been shown to act as chemical messengers between certain bacteria and their hosts, triggering cell signaling events unrelated to innate immune clearance [56–58]. For example, anhydromuropeptide release has been shown to play a critical role in enabling the bacterium *Vibrio fischeri* to establish its symbiosis with the squid *Euprymna scolopes.* During growth, *V. fischeri* releases 1,6-anhydro-disaccharide-tetrapeptides that help trigger key morphogenic events in the squid's nascent light organ, ultimately facilitating its colonization by *V. fischeri* [59]. By analogy, it is possible that 1,6-anhydro-disaccharide-peptides shed by *B. burgdorferi* during its growth in the midgut of a feeding tick may alter the tick vector's physiology to facilitate *B. burgdorferi* colonization or subsequent transmission. Further experimentation will be required to test this hypothesis. Using the *bb0259* knockout (Δ*mltS*) strain, which sheds little to no anhydromuropeptides (Fig 4), may seem attractive for such studies. However, the Δ*mltS* strain is already expected to be non-infectious due to its cell motility defect [36]. It has been well-established that motility is critical for *B. burgdorferi*'s life cycle [60–62]. It is also possible that the release of PG^Bb fragments during *B. burgdorferi* growth does not confer any fitness advantage and is merely a consequence of the loss of PG recycling genes during the evolution of this spirochete as an obligate parasite, which has led to a reduction in genome size. *E. coli* mutants deficient for PG recycling have no reported fitness cost in laboratory cultures [56]. Furthermore, while *B. burgodorferi* lacks the AmpG transporter (which is specific to PG fragments containing an anhydro-$N$-acetylmuramyl residue [63]), it carries other transporters that allows this pathogen to uptake precursors of PG synthesis (such as Glc$N$Ac and amino acids) from its host environment [64–66]. This scavenging capability may obviate the need for a functional PG recycling pathway.

Importantly, our findings suggest that the pathogenic form of peptidoglycan in Lyme arthritis patients is polymeric PG^Bb material from dead spirochetes. In a study investigating *B. burgdorferi* burden and viability, all spirochetes identified in Lyme arthritis joint fluid samples were found to be moribund or dead, and spiking joint fluid samples (diluted 1:10 or 1:100) with $10^2$, $10^4$, or $10^6$ cultured *B. burgdorferi* cells was found to result in rapid spirochete killing [67]. Thus, the joint fluid, which contains high-titer *B. burgdorferi* antibodies, proinflammatory cytokines/chemokines, and complement components

[68,69], constitutes a generally lethal environment for spirochetes, thereby restricting infection to protected niches within the affected joint tissues [67]. We hypothesize that immunogenic MurNAc-peptides from the PG$^{Bb}$ sacculus become exposed to the immune system as *B. burgdorferi* cells lyse from the effects of initial immune responses or antibiotic treatment. Systemic injection of streptococcal PG in rats have shown that cell wall material can persist for weeks to months in the animals and that the localized presence of PG in joint tissues is associated with acute and chronic inflammation [70,71]. A study published during the revision of our manuscript demonstrates that polymeric PG$^{Bb}$ persists in the joints of Lyme arthritis patients [72], consistent with PG$^{Bb}$ from dead spirochetes being the pathogenic form.

In tissue culture of synovium from a patient with postinfectious Lyme arthritis, HLA-DR-expressing fibroblast-like synoviocytes, the most common cells in the synovial lesion, were found to be IFNγ-inducible antigen-presenting cells that present autoantigens [14]. Moreover, when cultures were primed with IFNγ and PG$^{Bb}$, they secreted much higher levels of IFNγ, suggesting that the combination of IFNγ and PG$^{Bb}$ may lead to enhanced activation of proinflammatory autoreactive T cells [14]. Several host factors may bolster responsiveness to PG-derived immunogens. For instance, increased NOD2 expression has been identified in synovial tissues isolated from patients with postinfectious Lyme arthritis [11]. In addition, the level of N-acetylmuramyl-L-alanine amidase activity and the abundance of PGLYRP2, which is expected to limit NOD2 activation, are low in synovial fluids—and thus presumably within the synovium— relative to serum (Figs 6 and S14). Low PGLYRP2 activity, combined with elevated NOD2 level, may predispose joint tissues to PG$^{Bb}$-mediated inflammation. Furthermore, we observed a large variability in PGLYRP2 and other PG-hydrolytic activities across samples (Fig 6). Whether this variability contributes to differences in disease progression among individuals is an intriguing possibility that will require longitudinal studies to be examined. It is also worth noting that PGLYRP2 activity has been reported to be particularly low in cerebrospinal fluid [51], raising the possibility that PG$^{Bb}$ may play a role in neuroborreliosis. Altogether, our study suggests a potential link between chronic inflammation and innate immunity.

Supplementing the inflamed tissues of Lyme arthritis patients with PGLYRP2 activity may, in theory, help mitigate PG-mediated immunopathology. Given the higher PGLYRP2 activity in the serum of Lyme arthritis patients (Fig 6), intraarticular injections of blood-derived products from autologous origin (i.e., obtained from the same individual) may be an effective treatment strategy for PG-induced pathology. For example, platelet-rich plasma injections are often used as orthobiologics to help heal damaged tissues [73,74]. Further studies on PGLYRP2 are recommended before exploring this treatment option for antibiotic-refractory Lyme arthritis and other potential PG-induced inflammatory pathologies [75–77], as PGLYRP2 may have pro-inflammatory activity downstream NOD2 [78].

## Materials and methods

### Bacterial strains and growth conditions

Strains used in this study are detailed in S2 Table. *B. burgdorferi* cells were cultured in complete Barbour-Stoenner-Kelly (BSK)-II medium in humidified incubators at 34ºC under 5% $CO_2$ atmosphere [64,79,80]. Complete BSK-II contained 50 g/L bovine serum albumin (BSA, Millipore # 81–003), 9.7 g/L CMRL-1066 (US Biological # C5900-01), 5 g/L neopeptone (Thermo Fisher # 211681), 6 g/L HEPES acid (Millipore # 391338), 5 g/L D-glucose (Sigma # G7021), 2 g/L yeastolate (Difco # 255772), 0.7 g/L sodium citrate (Sigma # C7254), 0.8 g/L sodium pyruvate (Sigma # P5280), 2.2 g/L sodium bicarbonate (Sigma # S5761) and 0.4 g/L N-acetyl-glucosamine (Sigma # A3286). The pH of the medium was adjusted to 7.6, then supplemented with 60 mL/L heat-inactivated rabbit serum. Heat inactivation involved incubating the serum at 50ºC in a water bath for 30 min prior to use in medium preparation. Unless explicitly noted, all *B. burgdorferi* cultures were maintained in exponential phase at a density below 5 x $10^7$ cells/mL. Cell density was determined through direct cell counting using disposable Petroff-Hausser chambers and darkfield microscopy as previously described [81]. When appropriate, relevant *B. burgdorferi* strains were cultured in the presence of antibiotics at the following final concentrations: 200 µg/mL

for kanamycin [82] (Sigma # K1377), 40 µg/mL for gentamicin [83] (Sigma # G1914), and 100 µg/mL for streptomycin [84] (Sigma # S6501).

## Verification of plasmid content in *B. burgdorferi* strains

Three biological replicates of *B. burgdorferi* strains B31 IR, K2, 297, and N40 were grown to a density between 1 and 4 x $10^7$ cells/mL in 14 mL complete BSK-II medium. The resulting cultures were centrifuged at 9000 x g for 5 min at 25ºC. Cell pellets were processed using a QIAGEN DNeasy Blood & Tissue Kit (# 69504) to isolate genomic DNA (gDNA). Purified gDNA was sent for Illumina whole-genome sequencing using SeqCenter in Pittsburgh, PA, USA with a minimum read count of 1.33 million reads per sample. Different reference genomes were used for analysis: ASM4079079v1 for strain 297 [85,86], ASM868v2 for strains B31 IR and K2 [16,85], and ASM16663v1 for strain N40 [85,86]. Since the reference genome ASM4079079v1 for strain 297 is missing five cp32 plasmids, additional sequences were obtained from a previous study [20] and appended to the reference: cp32–1 (Accession # CP002254), cp32–5 (Accession # CP002261), cp32–6 (Accession # CP002263), cp32–9 (Accession # CP002262), and cp32–12 (Accession # CP002255). Reads were aligned to their respective reference using Breseq [87]. The resulting alignment ".bam" files were then used to calculate coverage using the Python package HTSeq [88]. The coverage array was loaded using the function *GenomicArray* and then quantified using the function *GenomicInterval* for the number of reads in a 4000-bp search window across each genetic element. The counts in each search window were normalized to the total number of alignments for the genetic element in question. The fold change of each genetic element was calculated by taking the mean counts for each element and dividing it by the mean counts for the chromosome. Due to the repetitive nature of some *B. burgdorferi* plasmid sequences, plasmid elements with a fold change under the arbitrary value of 0.25 were counted as missing that plasmid.

## Verification of the Himar*1* transposon insertion site in the *bb0605*::Himar*1* strain T08TC493

*B. burgdorferi* strains 5A18NP1 (B31-A) and T08TC493 (5A18NP1 *bb0605*::Himar*1*) were grown to 5 x $10^7$ cells/mL in 14 mL complete BSK-II. They were then centrifuged at 9000 x g for 5 min at 25ºC. Cell pellets were then processed using a QIAGEN DNeasy Blood & Tissue Kit (# 69504) to isolate genomic DNA (gDNA). The *bb0605* locus was then amplified by PCR using primers 5'- AAATGGCTCCTTTTTAATTGTATTGTAAT -3' and 5'- TTCACCAGGAACTATTATTGTAACAT -3'. After confirming amplification by agarose gel electrophoresis, the remaining volume of PCR products was processed using a QIAquick PCR Purification Kit (QIAGEN # 28104) and then the PCR products were sequenced to determine the insertion site of the Himar*1* transposon, which was found to have disrupted the *bb0605* open reading frame 300 bp downstream of the first codon.

## Purification of whole *B. burgdorferi* PG sacculi

Sacculi were purified in a manner similar to that described previously [89]. Briefly, two flasks of 1 L BSK-II medium were inoculated with *B. burgdorferi* to a starting density of 1 x $10^4$ cells/mL. Once the cultures reached 1 x $10^8$ cells/mL (Figs 3 and 6 and S7, and S14), or after 10 days in stationary phase (S13 Fig), cells were centrifuged at 6000 x g for 20 min at 4ºC in successive 333 mL batches into the same two 500 ml polypropylene centrifuge tubes (Corning # 431123). Between each centrifugation, pellets were dislodged by vigorously swirling with the added culture. After the entire culture volume was centrifuged, the pellets were washed three times with cold phosphate-buffered saline (PBS, VWR # 76371–734), shaking vigorously to achieve a homogenous distribution of cells. The final pellets were stored at -80ºC overnight.

The following day, the pellets were resuspended in 6 mL chilled PBS and placed on ice. Separately, 6 mL of 10% w/v sodium dodecyl sulfate (SDS) per 1 L of cells to be processed was brought to a boil. The cell suspension in PBS was then added dropwise to the boiling SDS and allowed to continue boiling for an additional 30 min before being allowed to cool to room temperature. The lysed cell suspension was then transferred to glass culture tubes that were placed in a beaker

filled appropriately with Milli-Q H$_2$O. The glass culture tubes were mixed with a stir bar as the water was boiled for 2.5 h on a hotplate. The hotplate was subsequently turned off while the samples continued to be stirred overnight as they re-equilibrated to room temperature.

On the third day, the content of each tube was transferred to ultracentrifuge tubes and pelleted at 150,000 x g for 20 min at 20ºC. The pellets were washed four times with 10.5 mL Milli-Q H2O, centrifuging each time at 150,000 x g for 20 min at 20ºC. After washing, the pellets were resuspended in 1 mL 100 mM Tris-HCl buffer (pH 7.5). Ten microliters of a 20 µg/mL solution of alpha-amylase (Sigma # 10102814001) were added to each suspension and incubated at 37ºC for 2 h, shaking at 220 rpm. Next, 1 mL Tris-HCl buffer (pH 7.5), 40 µL 1 M MgSO$_4$, and 2 µ L nuclease mix [i.e., 100 µg/mL DNase I (Roche # 11284932001) + 500 µg/mL RNase A (Roche # 10109142001) dissolved in Milli-Q water] were added to each sample, followed by a 2 h incubation at 37ºC while shaking at 220 rpm. Lastly, 50 µL of 50 mM CaCl$_2$ and 100 µL of a 2 mg/mL solution of chymotrypsin (Sigma # C4129) were added to each sample. The final suspensions were allowed to shake overnight at 220 rpm and 37ºC.

On the final day, the chymotrypsin was inactivated by adding 200 µL of 10% SDS and boiling the samples in a water bath for 10 min. The contents of all tubes representing a common sample were combined into a single ultracentrifuge tube and washed six times with 10.5 mL Milli-Q H$_2$O. For each wash, the samples were centrifuged at 150,000 x g for 20 min at 20ºC in an ultracentrifuge. After the final wash, the pellet was resuspended in 1 mL sterile Milli-Q H$_2$O. Two hundred microliters of the final suspension were transferred to a pre-weighed Eppendorf tube for lyophilization. The dry mass of the lyophilized product was measured to estimate the PG concentration of the starting suspension.

**Extraction of PG$^{Bb}$ fragments from culture supernatants**

*B. burgdorferi* cultures at the relevant densities (between $10^4$ and $10^8$ cells/mL) were used (see Figs 1, 2, 4, and S1–S5, S7, and S9 for respective harvest times/cell densities). PG material was extracted using an acetonitrile:methanol extraction method. Briefly, 1 mL of relevant cultures was centrifuged and the supernatant filter-sterilized through a 0.1 µm PVDF syringe filter (Celltreat # 229740) into a separate tube, which was then stored at -20ºC until use. One hundred-fifty microliters of culture supernatant was mixed with 450 µL of a 1:1 mixture of acetonitrile:methanol (prepared fresh, chilled to -20ºC before use), vortexed, and sonicated at room temperature for 10 min in a water bath sonicator. The sample was then placed at -20ºC for 1 h and vortexed every 30 min. Insoluble debris were pelleted at 15,000 x g for 20 min at 4ºC, then ~85% of the supernatant volume was carefully transferred to a new tube. The solution was dried in a Labconco centrivap (#7810014) at 37ºC for ~6 h after which the pellets were stored at -20ºC until use. Samples were resuspended with 150 µL LC-MS grade water (MilliQ H$_2$O) for mass spectrometry.

**Digestion of *B. burgdorferi* sacculi for liquid chromatography coupled to mass spectrometry**

To determine the composition of PG$^{Bb}$ sacculi isolated from either the *bb0605*::Himar*1* (T08TC493) strain or its parent (5A18NP1) (Fig 3), muramidase reactions were prepared consisting of 2 µL of mutanolysin (1 mg/mL), 2 µL of Tris-HCl (pH 7), ~125 µg of PG, and Milli-Q H$_2$O to bring the final volume to 100 µL. These samples were vortexed, spun down, and then incubated overnight at 37ºC, shaking at 700 rpm. The following day, digests were boiled at 100ºC for 5 min to denature the mutanolysin. The samples were then centrifuged at 20,000 x g for 15 min at room temperature. The supernatant was diluted 1:10 in Milli-Q H$_2$O, with 150 µL being set aside for LC-MS. The same procedure was performed to analyze the composition of mutanolysin-digested PG$^{Bb}$ sacculi isolated from 10-day-old stationary phase cultures of the B31 IR strain (S13 Fig).

To assess the ability of joint fluids or sera from Lyme arthritis patients to digest PG, 5 µL whole sacculi (1.73 mg/mL) from strain T08TC493 was mixed with 90 µL of each body fluid and 5 µL Milli-Q H$_2$O. A negative control consisting only of 45 µL of each patient fluid and 5 µL Milli-Q H$_2$O was included. Half of the samples to which PG was added were immediately set aside to serve as a 0 h time point, while the rest of the samples was incubated at 37ºC for 6 h, shaking

at 750 rpm. These samples were then stored at -20ºC until they were ready for processing. Once ready, 40 µL thawed samples were mixed with 160 µL 1:1 mixture of acetonitrile:methanol. Following this, samples were processed further for LC-MS as described above.

To test whether components of the BSK-II medium degrade PG, 5 µL whole sacculi (1.25 mg/mL) purified from strain B31 IR was mixed with 5 µL Milli-Q $H_2O$ and 90 µL BSK-II medium (containing 6% of heat-inactivated rabbit serum) or 100% heat-inactivated serum. Replicates of rabbit serum were derived from separate lot numbers to check batch-to-batch variability. The "no PG^Bb" controls contained 45 µL of BSK II or heat-inactivated rabbit serum and 5 µL Milli-Q $H_2O$. Samples were incubated at 37ºC for 24 h, shaking at 750 rpm. Samples were then further processed for LC-MS as described above.

**Liquid chromatography coupled to mass spectrometry**

LC-MS experiments were performed using an Agilent QTOF 6545, coupled with an Agilent 1290 Infinity II ultra-high-pressure liquid chromatography (UHPLC) device. Samples were separated on a Kinetex Polar C18 column with a 100 Å pore size, 2.6 µm particle size and 2.1 x 100 mm in size (Phenomenex #00A-4759-AN), with the guard column Securityguard Ultra Holder (Phenomenex #AJ0–9000). Buffers used were A: LCMS-grade $H_2O$ + 0.1% formic acid (Sigma # 1590132500) and B: acetonitrile + 0.1% formic acid (Sigma # 900686) on a gradient from 0–95% B. The gradient for the LC experiments is described in S3 Table. The column temperature was maintained at 40ºC during the experiment. Ten microliters of sample were injected per run.

Quadrupole time-of-flight (QTOF) MS settings were as follows. All PG^Bb molecules were detected in the positive ionization mode with the instrument mass range set to 70–1700 Da. The scan rate was 1.5 spectra/s. Fragmentor voltage was set to 175 V. For MS/MS fragmentation (S1-S5 Figs), the QTOF collision energy for the targeted released muropeptide masses was set as indicated in S1–S5 Figs.

For analysis, Agilent ".d" files were converted into ".mzML" files using MSConvert [90]. The python package pymzmL was used to analyze the mzML files [91]. Extracted ion chromatograms were constructed using pymzmL's *run* function and by searching each scan in the retention time for the candidate molecule ± 20 parts per million (ppm), while the total ion chromatograms summed total counts across every run time. Integrated signal for a single species was performed by summing the relevant area under the ion peaks.

The initial screen (Fig 1C) was analyzed using the XCMS package in R [92–94]. Briefly, chromatogram peaks were detected using the function *findChromPeaks* with parameters defined by the function *CentWaveParam*. Peak criteria were defined as $ppm = 20$, $peakwidth = c(6, 30)$, $noise = 1000$, $prefilter = c(3, 1000)$, and $snthresh = 8$. Neighboring peaks were merged using the function *RefineChromPeaks*, specifying refining parameters with *MergeNeighboringPeaksParam*. Merging criteria were $expandRT = 0.1$, $ppm = 2.5$, and $minProp = 0.75$. Statistical significance was determined by XCMS's *pval* function, which uses a Welch's two-sample *t*-test. All significant hits ($p < 0.05$, fold change > 3) were then checked against the theoretical PG^Bb masses in python using the packages *numpy* and *pandas* [95,96]. Identified targets were validated based on enrichment of EIC peaks in culture supernatants over BSK-II alone, valid isotopic distributions, and MS/MS fragmentation.

To screen for putative PG species (Figs 6A and S14A), masses of theoretical PG^Bb fragment species were concatenated into a single vector containing all [M+H]+ and [M+H]2 + values. Extracted ion count (EIC) profiles were constructed for every value between retention times of 1 and 15 min of a 20-min run to exclude nonspecific compounds that wash off the column. Scipy's *find_peaks* function was then used to identify peaks as $prominence = 1000$, $width = (6, 30)$ in retention time seconds, $rel\_height = 1$ to capture width from the base of the peaks of interest, and $threshold = 7$ for a minimum signal-to-noise threshold. Identified hits were then verified to have an isotopic abundance distribution of two decaying peaks, the first of which being at least 1000 counts high. The most abundant peak was selected and summed for quantification. Identified masses from this process were then checked against the theoretical PG^Bb library. Identified hits were verified by constructing EICs profiles and checking that the elution profiles were consistent with previous observations. Inconsistent observations were removed from consideration.

Heat maps in Figs 6A and S14A were compiled from the extracted ion counts, keeping only features that showed at least three-fold enrichment over the negative controls (0 h or no-PG$^{Bb}$ conditions). The maximum value of extracted ions for features corresponding to identified PG$^{Bb}$ species in any sample had to be at least 10,000 to be considered.

## Microscopy

The cell densities of *B. burgdorferi* cultures were determined by dilution in PBS and using an In-Cyto C-Chip disposable hematocytometer (InCyto # DHC-N01) and darkfield illumination in a Nikon Eclipse E600 microscope with a 40x 0.55 NA Ph2 phase contrast air objective and darkfield condenser lens. For fluorescence microscopy, samples were spotted on 2% PBS agarose pads [27,97]; covered with a No. 1.5 coverslip; and sealed with VALAP, a 1:1:1 mixture of **Va**seline petroleum jelly (Amazon ASIN B07MD6HJT4), **la**nolin (Spectrum # LA109) and **p**araffin wax (Fisher # 18-607-738), on all edges. Specimens were imaged using Nikon Eclipse Ti inverted microscopes with a 100x Plan Apo 1.45 NA phase contrast oil objective, Hamamatsu Orca-Flash4.0 V2 CMOS camera, a Sola LE light source (phase contrast), and a Spectra X Light engine. For the acquisition of fluorescent HADA images a DAPI filter set was used (Chroma 49028: ET395/25 ex, dichroic T425lpxr, ET460/50 em). The microscope was controlled by NIS-Elements AR.

## HADA labeling of *B. burgdorferi* cells

Strains 5A18NP and T08TC493 were grown to high $10^6$ or low $10^7$ cells/mL densities. The morning of the experiment, 1 mL of each culture was set aside. Two microliters of 50 mM HADA (Tocris # 6647) were pipetted into each tube, which was then covered in aluminum foil and mixed by inversion. The samples were incubated at 34°C. After 1 h and 8 h of incubation, 200 µL was removed from the samples and placed in a 1.5 mL Eppendorf tube. These 200 µL aliquots were immediately centrifuged at 9,000 x g for 5 min at room temperature, washing two times in 500 µL BSK-II without phenol red. The final pellet was resuspended in 20–100 µL of phenol red-free BSK-II depending on the pellet size. Five microliters were spotted on a 2% PBS agarose pad for immediate imaging. Phase contrast images were acquired with a 500 ms exposure while HADA fluorescence was imaged with the DAPI filter set using 385 nm excitation at 40% power with an exposure of 1000 or 250 ms based on the 1 or 8 h of incubation time with HADA, respectively. For analysis, the intensity of cells was multiplied to equalize all time points to 1000 ms for comparison.

## Image analysis

FIJI [98] was used for the preliminary assessment of fluorescence images, then Python was used for the remaining work. First, to facilitate analysis, we loaded large Nikon ND2 files as arrays using the previously published function *nd2_to_array* function [99]. To create reliable masks of phase contrast images, an adaptive threshold from the Python package OpenCV (*cv2.adaptiveThreshold*) was applied, excluding objects smaller than 500 square pixels. Any surviving masks were kept and skeletonized. Cell skeletons were used to measure the width along the mask. Cells at least 4 µm long and cell masks less than 0.7 µm wide were selected. The surviving masks were assigned an ID and archived in a Pandas data frame for cell length and fluorescence measurements.

Linescans were generated by first measuring the medial axis down the center of the mask using the function *get_medial_axis* generated as before [100]. The medial axis is a sub-pixel polynomial fitting from pole-to-pole. The medial axis was then used to measure cell lengths and retrieve the corresponding pixel intensities in the fluorescence image. The medial axis, cell length, and linescan measurements were also saved in the same Pandas data frame corresponding to each cell ID. Demographs were created by constructing a Python dictionary of all cell linescans in a dataset. They were ordered from shortest to longest, then arranged into a numpy array centered at mid-cell. Each line scan was normalized by z-score to show where signal was enriched.

To measure fluorescence in each cell, background-subtracted images were produced in Python using a modified background subtraction algorithm [99]. Phase contrast images were segmented first using OpenCV's adaptive threshold

(*cv2.adaptiveThreshold*) to remove all potential fluorescing objects. These masks were dilated using Scipy's *binary_dilation* function with ten iterations [101]. The dilated objects were then removed from the fluorescent image. Surviving pixels were binned and averaged using a rolling window across the *x* and *y* dimensions, then smoothed with a Gaussian kernel to obtain an estimated "background-only" image. Cell masks were used to measure the mean intensity (integrated intensity per unit pixel area) from every cell in the background-subtracted images.

## Human subject samples

The study patients with Lyme arthritis met the criteria of the Centers for Disease Control and Prevention for *B. burgdorferi* infections [102]. They had knee swelling and pain and positive antibody responses to *B. burgdorferi*, as determined by ELISA and Western blot. The patients were treated according to the guidelines of the Infectious Diseases Society of America [103]. They had persistent arthritis despite one to two months of oral antibiotic therapy (usually doxycycline), followed by an additional one month of intravenous (IV) antibiotic therapy (ceftriaxone).

The six Lyme disease patients had a range of inflammatory responses and post-treatment durations of arthritis that were typical of more severe Lyme arthritis. Three patients had the resolution of arthritis one to three months after IV antibiotic therapy, whereas the other three had postinfectious arthritis lasting four to nine months after IV therapy. In three of the six patients, synovial fluid samples were obtained prior to starting IV antibiotic therapy when joints are typically swollen. In the remaining three patients, synovial fluid samples were obtained prior to starting synovectomy (n = 1) or therapy with disease-modifying antirheumatic drugs or DMARDs (n = 2). When samples were obtained, the median white cell count in synovial fluids was 17,022 cells/mm$^3$ (range 1,567–31,286); whereas in blood, the median C-reactive protein (CRP) value was 28.4 (range 8–71.4), and the median erythrocyte sedimentation rate (ESR) was 28.5 (range 7–49). In four of the six patients, corresponding serum samples were available.

The three patients with rheumatoid arthritis had symmetrical polyarthritis affecting large and small joints. Samples were obtained during flares of arthritis one, four, and six years after disease onset. Patients were taking disease modifying anti-rheumatic drugs (DMARDs) at that time. Synovial fluid white cell counts ranged from 7,776–20,300 cells/mm$^3$, ESR values were high (62–66), and CRP values were also elevated (18.5 to 27.8). One patient had positive test results for both anti-citrullinated protein antibodies (ACPA) and rheumatoid factor (RF), one had a positive test for ACPA alone, and one had negative results for both tests.

To exclude sampling artifacts as an additional source of variation, serum and joint fluid samples were processed within hours after aspiration of synovial fluid. The samples were centrifuged at 300 x g for 10 min, followed by another centrifugation at 3000 x g for 10 min to remove cells and cell debris as previously described [104]. Samples were stored in 1.5 mL aliquots in small plastic screw-top tubes and frozen at -80ºC without dilution or preservatives on the same day synovial fluid was obtained. Before each use, samples were centrifuged at 2,000 x g for 1 min to pellet potential remaining debris. No sample had undergone more than two freeze-thaw cycles before use in this study.

## Enzyme-linked immunosorbent assays

Enzyme-linked immunosorbent assays (ELISA) were carried out using the PGLYRP2 ELISA Kit (MyBioSource # MBS2024542) and the Human Lysozyme ELISA Kit (Abcam # ab267798) using the manufacturer's protocol. The patient samples for PGLYRP2 detection were diluted 200-fold in PBS to fall within the dynamic range of the kit standards, while the samples for lysozyme detection were diluted 12,000-fold in PBS.

## THP-1 and NOD2 reporter cell stimulation

The panel of synthetic PG$^{Bb}$ molecules used for the immunological assays were synthesized as previously described [41]. Two assays were performed to examine the immunostimulatory activities of these molecules. In the first one, THP-1 cells (ATCC # TIB-202) were grown in a humidified incubator at 37ºC under a 5% $CO_2$ atmosphere in complete RPMI (cRPMI)

containing 10% fetal bovine serum (FBS) + 1% Pen-Strep (Gibco RPMI 1640 # 11875119; R&D systems FBS – premium select, heat inactivated # S11550H; Gibco Pen-Strep # 15140122), with cell densities maintained between $1 \times 10^5$ and $1.5 \times 10^6$ cells/mL. For $PG^{Bb}$ stimulation experiments, THP-1 cells were differentiated into macrophage-like cells in the presence of phorbol 12-myristate 13-acetate (PMA, InvivoGen # tlrl-pma). More specifically, cells were diluted to $1.25 \times 10^6$ cells/mL in cRPMI supplemented with 40 nM PMA, and 200 µL of the resulting cell suspension were added to the appropriate wells of a tissue culture-treated, flat-bottom 96-well plate (Fisher Scientific # FB012931) for a total of $2.5 \times 10^5$ cells/well. After 48 h, the PMA-containing medium was removed by aspiration, 200 µL of fresh cRPMI lacking PMA was added to each well, and the cells were allowed to rest for 72 h. Following this rest period, the medium in each well was exchanged for serum-free RPMI and the cells were starved in this manner for 4 h prior to $PG^{Bb}$ stimulation. At the end of this interval, the serum-free medium was removed from each well using an aspirating pipette and then replaced with cRPMI containing a synthetic $PG^{Bb}$ fragment at a final concentration of 30 µM. To account for differences in stock concentrations, each of the different potential ligands tested was first diluted in ultrapure water (American Bio # AB02123) to a concentration of 1 mM prior to further dilution to 30 µM in cRPMI. After a 24-h stimulation period, the supernatant was harvested from each well and immediately frozen at -80ºC. The supernatants were subsequently used for cytokine profiling by Luminex assay performed at Eve Technologies.

For the NOD2 reporter assay, HEK-Blue NOD2-overexpressing (hNOD2) and NOD2-deficient (Null2) cells (InvivoGen # hkb-hnod2v2 and hkb-null2, respectively) were maintained according to the manufacturer's recommendations. In brief, both cell lines were grown in Dulbecco's modified Eagle's medium (Sigma-Aldrich # D5796) containing 4.5 g/L glucose and 2 mM L-glutamine, and further supplemented with 10% FBS, 1% Pen-Strep, and 100 µg/mL Normocin (InvivoGen # ant-nr). Selective antibiotics were used in culturing both cell lines, beginning with the third passage after revival from frozen stock. Blasticidin (30 µg/mL, InvivoGen # ant-bl) and Zeocin (100 µg/mL, InvivoGen # ant-zn) were added to hNOD2 cell cultures, while only Zeocin was used for Null2 cell cultures. For $PG^{Bb}$ stimulation and measurement of the resulting NOD2-dependent NF-κB activation by secreted embryonic alkaline phosphatase (SEAP) assay, 20 µL of each synthetic PG molecule was prepared by dilution in ultrapure water to a stock concentration of 300 µM and added to the appropriate wells of a tissue culture-treated, flat-bottom 96-well plate to the indicated final concentrations. The hNOD2 and Null2 cells were detached from the T75 flasks in which they had been expanded by exchanging the growth medium in each for 5 mL pre-warmed PBS and gently tapping the flask sides. Cells were fully resuspended by gentle pipetting and then counted using a disposable hemocytometer. Pre-warmed HEK-Blue Detection medium (InvivoGen # hb-det2; reconstituted according to manufacturer's recommendations) was then added to each cell suspension to achieve a final cell density of $2.8 \times 10^5$ cells/mL. Next, 180 µL of the resulting cell suspension (or a total of approximately $5 \times 10^4$ cells) were added to the appropriate wells of the 96-well plate described above such that the final PG fragment concentration in each well was either 1 or 30 µM. The plate was then placed at 37ºC in a humidified incubator containing 5% $CO_2$ and SEAP production was assessed by absorbance measurement at 655 nm (Molecular devices, SpectraMax i3X) after 16 and 22 h of incubation.

For S11D Fig, about 50,000 HEK-Blue hNOD2 reporter cells (eighth passage) in a volume of 180 µL HEK-Blue Detection medium were added to each well containing 20 µL synthetic MDP (Invivogen # tlrl-mdp) diluted in ultrapure water such that the final MDP concentration ranged from 5 nM to 100 µM across a row of a 96-well plate. Plates were incubated for 16 h in a humidified incubator at 37ºC and 5% $CO_2$, with subsequent absorbance measurements being obtained at 655 nm.

To prepare *B. burgdorferi* culture supernatants for HEK-Blue hNOD2 reporter stimulation (S12B Fig), *B. burgdorferi* starter cultures were used to inoculate 10 mL complete BSK-II medium (supplemented with 200 µg/mL kanamycin) to obtain a starting cell density of ~$10^4$ cells/mL. As a medium-only control, 10 mL complete BSK-II medium (also supplemented with kanamycin) was subjected to the same conditions (humidified 34ºC and 5% $CO_2$ environment with tube cap loosened) as *B. burgdorferi* cultures. Once the cultures reached a spirochete density of ~$10^7$ cells/mL, cells were pelleted, and supernatants transferred to fresh tubes that were immediately stored at -80ºC. Prior to application to HEK-Blue hNOD2-expressing reporter cells (fifth passage), thawed supernatants were passed through a 0.1 µm PES membrane filter (Foxx Life Sciences # 146–4113-RLS). HEK-Blue reporter cells were stimulated as before by exposure to 20 µL *B.*

*burgdorferi* culture supernatant, complete BSK-II medium alone, MDP dissolved in ultrapure water (1 µM final concentration), or ultrapure water alone.

## Computational analyses

For all custom-made analyses, R and Python scripts are available on the Jacobs-Wagner lab Github (https://github.com/JacobsWagnerLab/published/tree/master/McCausland_et_al_2025). Python packages numpy, scipy, scikit-image, seaborn, and pandas used in this study have previously been described [95,96,101,105,106]. The R package XCMS was previously described [92].

   Raw MS data are available in the GlycoPOST repository [107] under accession numbers GPST000537 and GPST000595. Raw microscopy images are available in the BioImage Archive [108] under the accession number S-BIAD1573.

## Supporting information

**S1 Fig. Fragmentation of PG<sup>Bb</sup> fragment ʟ-Ala-ᴅ-Glu-ʟ-Orn(Gly)-ᴅ-Ala-ᴅ-Ala. A.** Schematic outlining each amino acid in ʟ-Ala-ᴅ-Glu-ʟ-Orn(Gly)-ᴅ-Ala-ᴅ-Ala, along with notations of the origin of each identified fragment after MS/MS. **B.** EIC profile of the identified [M+H]+ profile. The peak of interest is shaded in red. **C.** The MS2 spectrum at the run time where the EIC peak in (B) is at maximum intensity. The collision energy used to fragment this molecule is indicated, and each identified fragment is marked with a letter corresponding to the schematic in (A).
(PDF)

**S2 Fig. Fragmentation of PG<sup>Bb</sup> fragment ʟ-Ala-ᴅ-Glu-ʟ-Orn-ᴅ-Ala-ᴅ-Ala. A.** Schematic outlining each amino acid in ʟ-Ala-ᴅ-Glu-ʟ-Orn-ᴅ-Ala-ᴅ-Ala, along with notations of the origin of each identified fragment after MS/MS. **B.** EIC profile of the identified [M+H]+ profile. The peak of interest is shaded in red. **C.** The MS2 spectrum at the run time where the EIC peak in (B) is at maximum intensity. The collision energy used to fragment this molecule is indicated, and each identified fragment is marked with a letter corresponding to the schematic in (A).
(PDF)

**S3 Fig. Fragmentation of PG<sup>Bb</sup> fragment Glc*N*Ac-AnhMur*N*Ac-ʟ-Ala-ᴅ-Glu-ʟ-Orn(Gly)-ᴅ-Ala-ᴅ-Ala. A.** Schematic outlining the sugars and amino acids in Glc*N*Ac-AnhMur*N*Ac-ʟ-Ala-ᴅ-Glu-ʟ-Orn(Gly)-ᴅ-Ala-ᴅ-Ala, along with notations of the origin of each identified fragment after MS/MS. **B.** EIC profile of the identified [M+H]+ profile of peak 1, shaded in red. **C.** The MS2 spectrum at the run time where EIC peak 1 in (B) is at maximum intensity. **D.** EIC profile of the identified [M+H]+ profile of peak 2, shaded in red. **E.** The MS2 spectrum at the run time where EIC peak 2 in (D) is at maximum intensity. **F.** EIC profile of the identified [M+H]+ profile of peak 3, shaded in red. **G.** The MS2 spectrum at the run time where EIC peak 3 in (F) is at maximum intensity. For (C), (E), and (G), the collision energies used to fragment this molecule are noted in the titles, and each identified fragment is marked with a letter corresponding to the schematic in (A).
(PDF)

**S4 Fig. Fragmentation of PG<sup>Bb</sup> fragment Glc*N*Ac-AnhMur*N*Ac-ʟ-Ala-ᴅ-Glu-ʟ-Orn(Gly). A.** Schematic outlining the sugars and amino acids in Glc*N*Ac-AnhMur*N*Ac-ʟ-Ala-ᴅ-Glu-ʟ-Orn(Gly), along with notations of the origin of each identified fragment after MS/MS. **B.** EIC profile of the identified [M+H]+ profile. The peak of interest is shaded in red. **C.** The MS2 spectrum at the run time where the EIC peak in (B) is at maximum intensity. The collision energy used to fragment this molecule is indicated, and each identified fragment is marked with a letter corresponding to the schematic in (A).
(PDF)

**S5 Fig. Fragmentation of PG<sup>Bb</sup> fragment Glc*N*Ac-AnhMur*N*Ac. A.** Schematic outlining each sugar in Glc*N*Ac-AnhMur*N*Ac, along with notations of the origin of each identified fragment after MS/MS. **B.** EIC profile of the identified [M+H]+ profile. The peak of interest is shaded in red. **C.** The MS2 spectrum at the run time where the EIC peak in (B) is

at maximum intensity. The collision energy used to fragment this molecule is indicated, and each identified fragment is marked with a letter corresponding to the schematic in (A).
(PDF)

**S6 Fig. Plasmid verification in *B. burgdorferi* strains used in the study using whole-genome sequencing.** Fold change over mean chromosome read counts for each plasmid present in strains: **A.** B31 IR, **B.** K2, **C.** N40, and **D.** 297. For each strain, a fold change near 0 (< 0.25) indicates that it is missing that plasmid.
(PDF)

**S7 Fig. *N*-acetylmuramyl-L-alanine amidase activity present in BSK-II and heat-inactivated rabbit serum.** BSK-II or heat-inactivated rabbit serum was incubated at 37ºC in the presence or absence of purified PG$^{Bb}$ sacculi for 24 h prior to LC-MS. **A.** Schematic of *N*-acetylmuramyl-L-alanine amidase activity when mixed with purified PG$^{Bb}$. Cut sites by a *N*-acetylmuramyl-L-alanine amidase are shown by red curvy lines. **B.** Digestion in BSK-II. **C.** Digestion in heat-inactivated rabbit serum (100%). For **B-C**, plots show the extracted ion count (EIC) for the L-Ala-D-Glu-L-Orn(Gly), the predominant expected digestion product of PG$^{Bb}$ sacculi by a *N*-acetylmuramyl-L-alanine amidase. For both panels, the bar shows the mean and the dots represent the data of two biological replicates of BSK-II media and rabbit sera sourced from different lot numbers.
(PDF)

**S8 Fig. Structural prediction comparison between *B. burgdorferi* BB0605 and *E. coli* DacA. A.** Predicted structure of *B. burgdorferi* BB0605 using Foldseek [109]. **B.** Structure of *E. coli* DacA obtained from the Protein Data Bank (PDB) [110,111]. **C.** Alignment of BB0605 and DacA, performed using ChimeraX [112]. The root mean square deviation (RMSD) from alignment is presented, along with the alignment algorithm used.
(PDF)

**S9 Fig. Assessment of Mur*N*Ac-containing PG species in culture supernatants of *B. burgdorferi* strain K2.** Plots showing the extracted ion counts of Glc*N*Ac-Mur*N*Ac, Glc*N*Ac-Mur*N*Ac-L-Ala-D-Glu-L-Orn(Gly), and Glc*N*Ac-Mur*N*Ac-L-Ala-D-Glu-L-Orn(Gly)-D-Ala-D-Ala in culture supernatants (Sup) compared to medium alone (BSK).
(PDF)

**S10 Fig. Structural prediction comparison between *B. burgdorferi* BB0259 and *E. coli* Slt70. A.** Predicted structure of BB0259 using Foldseek [109]. **B.** Structure of *E. coli* Slt70 obtained from the Protein Data Bank (PDB) [111,113]. **C.** Alignment of BB0259 and Slt70 using ChimeraX [112]. The root mean square deviation from alignment is presented, along with the alignment algorithm used.
(PDF)

**S11 Fig. Stimulation of THP-1 and hNOD2 reporter cells. A.** Comparison between Mur*N*Ac-containing and AnhMur*N*Ac-containing species in Fig 5A. The schematic of the PG$^{Bb}$ species in each group is shown, along with a legend that defines each chemical moiety. Dots represent the means of each PG$^{Bb}$ compound. Bar heights represent the means for each compound group. The groups were compared using a Welch's *t*-test to account for different standard deviations and N values. **B.** Plot showing IL-8 production in differentiated THP-1 cells in the presence of the indicated PG$^{Bb}$ fragments. The error bars represent standard deviation of the mean, and bar height represents the mean. Dots represent data from three biological replicates. **C.** Comparison of Mur*N*Ac-containing and AnhMur*N*Ac-containing species in (B), including MDP as in (A). Dots represent the means of each PG$^{Bb}$ compound. Bar heights represent the means for each compound group. The groups were compared using a Welch's *t*-test to account for different standard deviations and N values. **D.** Dose-response curve of hNOD2 reporter cells to MDP in MilliQ H$_2$O. The line connects the mean measurements of each concentration, and the dots represent technical replicates.
(PDF)

**S12 Fig. Absence of detectable stimulation of hNod2 reporter cells by culture supernatants of various *B. burgdorferi* strains. A.** NOD2 data from Fig 5B with PG[Bb] grouped based on the presence of a Mur*N*Ac or AnhMur*N*Ac, as shown by the schematics and the legend. Dots represent the means of each PG[Bb] compound. Bar heights represent the means for each compound group. The lines above the plot show pairwise comparisons between each group using Welch's *t*-tests to account for different standard deviations and N values. The resulting *p*-values were adjusted using a Bonferonni correction for multiple comparisons. **B.** Plot showing the SEAP activity of hNOD2 reporter cells (used after fifth passage) following 16-h exposure to 1 µM MDP (positive control) compared to complete BSK-II medium (negative control), or supernatants of cultures in stationary phase for three days. Each dot is a technical replicate, and the height of each bar represents the mean.
(PDF)

**S13 Fig. Ion count profiles of Mur*N*Ac- and AnhMur*N*Ac-containing monomers in PG[Bb] sacculi isolated from 10-day-old stationary phase cultures of *B. burgdorferi*.** PG[Bb] sacculi were isolated from 10-day-old stationary phase cultures of B31 IR cells and digested with mutanolysin. Digest products were then analyzed by LC-MS. **A.** Extracted ion count profiles for Glc*N*Ac-Mur*N*Ac-L-Ala-D-Glu-L-Orn(Gly) (blue) vs. Glc*N*Ac-AnhMur*N*Ac-L-Ala-D-Glu-L-Orn(Gly) (orange). **B.** Extracted ion count profiles for Glc*N*Ac-Mur*N*Ac-L-Ala-D-Glu-L-Orn(Gly)-D-Ala-D-Ala (blue) vs. Glc*N*Ac-AnhMur*N*Ac-L-Ala-D-Glu-L-Orn(Gly)-D-Ala-D-Ala (orange).
(PDF)

**S14 Fig. Patient joint fluid and serum digestions of purified PG[Bb] sacculi.** This heatmap is similar to Fig 6A except that negative control are samples in which no sacculi were added. Samples were incubated with MilliQ $H_2O$ (- PG[Bb] sacculi) or with PG[Bb] sacculi (+ PG[Bb] sacculi) for 6 h, then the resulting reaction products were analyzed by LC-MS. PG[Bb] species were detected by their predicted [M+H] value (S1 Dataset). All samples were derived from Lyme arthritis patients whose identities (IDs) are color-coded as in Fig 6A. The accompanying table contains the key to interpret the PG[Bb] fragment species in the heatmap. **B.** Representative extracted ion counts for peptide or sugar-peptide conjugate digestion products in serum and joint fluid samples predicted based on their masses. Each dot was derived from integrating the relevant EIC peak. These are the same fragments and analysis as in Fig 6B, but with a linear scale on the y-axis.
(PDF)

**S1 Table. Common PG species referenced within this manuscript.** The letter key refers to the heatmap table in Fig 6A. The peptidoglycan fragment masses were generated by summing the masses of each individual subunit and subtracting the mass of water with each addition.
(PDF)

**S2 Table. *B. burgdorferi* strains used in this study.**
(PDF)

**S3 Table.** The LC gradient for LC-MS experiments. Buffers A and B are noted in Materials and Methods.
(PDF)

**S1 Dataset.** Theoretical masses for monomeric and dimeric species of B. burgdorferi peptidoglycan. A mass library representing all permutations of possible *B. burgdorferi* peptidoglycan monomers and dimers was computationally generated by adding the mass of individual amino acid and saccharides and subtracting a water mass (18.0105647) for each addition. Each addition or subtraction of one proton used the mass of hydrogen (1.007276), and the mass of the sodium adduct corresponds to 22.98976928. See the tab "Muropeptide_letter_key" for a letter definition of the name listed in the "species" column.
(XLSX)

## Acknowledgments

We would like to thank Drs. Jonathon Z. Long, Veronica L. Li, Yuqin Dai, and Mengying Fu for their guidance on the mass spectrometry experiments. We also thank the Nucleus at the Stanford Sarafan ChEM-H insititute for access to mass spectrometry equipment. We are also grateful to the Jacobs-Wagner Laboratory for support, discussion, and critical reading of the manuscript. We would also like to thank the members of the Rego and Bockenstedt laboratories at Yale School of Medicine for providing lab space, equipment, and expertise to Z.A.K. to facilitate this work. Portions of this work were originally part of the dissertation of one of the co-first authors (Z.A.K.).

## Author contributions

**Conceptualization:** Zachary A. Kloos, Irnov Irnov, Christine Jacobs-Wagner.

**Data curation:** Joshua W. McCausland, Irnov Irnov.

**Formal analysis:** Joshua W. McCausland, Irnov Irnov.

**Funding acquisition:** Catherine L. Grimes, Christine Jacobs-Wagner.

**Investigation:** Joshua W. McCausland, Zachary A. Kloos, Irnov Irnov, Nicole D. Sonnert, Junhui Zhou, Rachel Putnik, Elizabeth A. Mueller.

**Methodology:** Irnov Irnov.

**Project administration:** Christine Jacobs-Wagner.

**Resources:** Allen C. Steere, Catherine L. Grimes.

**Software:** Joshua W. McCausland.

**Supervision:** Noah W. Palm, Catherine L. Grimes, Christine Jacobs-Wagner.

**Validation:** Joshua W. McCausland.

**Visualization:** Joshua W. McCausland, Irnov Irnov.

**Writing – original draft:** Joshua W. McCausland, Christine Jacobs-Wagner.

**Writing – review & editing:** Joshua W. McCausland, Zachary A. Kloos, Irnov Irnov, Nicole D. Sonnert, Junhui Zhou, Rachel Putnik, Elizabeth A. Mueller, Allen C. Steere, Noah W. Palm, Catherine L. Grimes, Christine Jacobs-Wagner.

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
