## [Decision Letter · Decision Letter 0]

PPATHOGENS-D-25-00051

Bacterial and host enzymes modulate the pro-inflammatory response elicited by the peptidoglycan of Lyme disease agent Borrelia burgdorferi

PLOS Pathogens

Dear Christine,

Thank you for submitting your manuscript to PLOS Pathogens. The study described is very interesting and carefully performed. However, the reviewers had some comments to enhance the manuscript. Therefore, we invite you to submit a revised version of the manuscript that addresses the points raised during the review process.

Please submit your revised manuscript within 30 days Apr 06 2025 11:59PM. If you will need more time than this to complete your revisions, please reply to this message or contact the journal office at plospathogens@plos.org. Please include the following items when submitting your revised manuscript:

We look forward to receiving your revised manuscript.

Kind regards,

John

John M Leong

Academic Editor

PLOS Pathogens

D. Scott Samuels

Section Editor

PLOS Pathogens

Sumita Bhaduri-McIntosh

Editor-in-Chief

PLOS Pathogens

orcid.org/0000-0003-2946-9497

Michael Malim

Editor-in-Chief

PLOS Pathogens

orcid.org/0000-0002-7699-2064

**Journal Requirements:**

At this stage, the following Authors/Authors require contributions: Joshua W. McCausland, Zachary A. Kloos, Irnov Irnov, Nicole D. Sonnert, Junhui Zhou, Rachel Putnik, Elizabeth A. Mueller, Allen C. Steere, Noah W. Palm, Catherine L. Grimes, and Christine Jacobs-Wagner. Please ensure that the full contributions of each author are acknowledged in the "Add/Edit/Remove Authors" section of our submission form.

3) Please ensure that the funders and grant numbers match between the Financial Disclosure field and the Funding Information tab in your submission form. Note that the funders must be provided in the same order in both places as well. State what role the funders took in the study. If the funders had no role in your study, please state: "The funders had no role in study design, data collection and analysis, decision to publish, or preparation of the manuscript.".

**Reviewers' Comments:**

Reviewer's Responses to Questions

**Part I - Summary**

Reviewer #1: This extremely well executed but complicated study is one of a number from the Jacobs-Wagner group addressing fundamental aspects of Borrelia burgdorferi cell biology, in this case, turnover of peptidoglycan (PG) and release of PG fragments to the extracellular milieu. What makes their work unique is their ability to relate basic biochemical, structural, and genetic analysis of B. burgdorferi PG composition and turnover to the immunopathogenesis of chronic Lyme arthritis –one of the least well understood clinical manifestations of Lyme disease. The starting point is their use of advanced mass spectroscopy approaches to characterize PG fragments B. burgdorferi that accumulate in the culture medium during in vitro growth. They accomplish this using a novel approach – matching mass spec signatures to a computational library of PG monomers and dimers. They then go on to identify two key borrelial enzymes (BB0605, a carboxypeptidase renamed DacA, and BB0259, a lytic transglycosylase renamed MltS) involved in generating the fragments, and they demonstrate that fragments are generated primarily at sites of PG synthesis during spirochete replication. The unexpected finding (at least for this reviewer) is that the sugar containing fragments have an anhydro chemical modification that renders them relatively invisible to human PG innate sensing mechanisms (e.g., NOD2), a result that conflicts with their previously published findings. They address this discrepancy by showing that PG hydrolytic enzymes in synovial fluid can release proinflammatory PG fragments when incubated with PG sacculi, leading them to propose that post-treatment Lyme arthritis results from breakdown of PG sacculi from dead spirochetes acting as immunostimulants/adjuvants to drive autoimmune responses.

Without question, this work is a technical tour de force that deals with PG synthesis and turnover in B. burgdorferi at an unprecedented depth and with analytic and computational methodologies new to the field. Moreover, it can be argued that the emphasis in the current work on PG fragments derived from dead spirochetes makes sense given that chronic Lyme arthritis occurs following therapy when live organisms (indeed, even DNA detectable by PCR) are no longer present. Moreover, these new results raise interesting new questions about the balance between inflammation and immune evasiveness during B. burgdorferi infection that should stimulate work in the field. On the other hand, the work has two interrelated weaknesses. One is the not entirely convincing explanation for the discrepancy between the culture supernatant experiments here and reported previously. Nevertheless, it should be noted that the authors are to be commended for their forthright acknowledgement of this discordance and their willingness to set the record straight. The other is that it adds to the experimental mishaps that have marred efforts over the years to elucidate the pathogenesis of chronic Lyme arthritis. Given the complexity of the phenomenon and the fact that causal mechanisms cannot be pursued in the murine model, one hopes that these new findings set this line of investigation on a steady course.

Reviewer #2: The manuscript by McCausland et al. reports on the interplay between Borrelia burgdorferi peptidoglycan and the immune response. The paper details that PG from dead Borrelia is more immunogenic than that extruded from live bacteria, identifies two key enzymes involved in PG turnover in the spirochete pathogen, and suggests that variation in a human PG degradation enzyme could explain both interpersonal and infection niche-level variation in disease outcomes. The study is well-designed and well-written (in fact, I truly enjoyed reading it), and yields impactful insight into the molecular mechanisms underlying Lyme disease progression. All controls are in place, and the conclusions are sound, and supported by the data. I only have very minor comments on the text.

Reviewer #3: The questions posed by the authors of this manuscript are significant in Bb immunopathogenesis, particularly the question of whether the immunogenic properties of Bb-PG are different against live vs dead organisms. The authors show that Bb MltS (BB0259) cleaves Bb-PG fragments, namely into 1,6-anhydromuramyl-contiaining fragments which are poor inducers of NOD2 immune responses. On the host side, the activity of PGLYRP2, an enzyme that cleaves Bb-PG to render it less immunogenic, is lower in joint fluids compared to serum, a finding that is consistent with previous clinical findings of Bb-PG-induced joint arthritis.

Major concerns focus on the conclusions drawn from the human patient samples and Bb-PG produced by “dead” vs. “live” Bb. Overall, this manuscript contains well-controlled experiments that would be helpful to those studying Bb immunopathogenesis from both host and bacteria perspectives.

**Part II – Major Issues: Key Experiments Required for Acceptance**

Reviewer #1: No major issues.

Reviewer #2: none

Reviewer #3: • The scientific question of comparing Bb-PG from live vs dead Bb is of great interest. While the authors cite data from Li et al. 2011 (discussed in lines 448-451) which demonstrates that Bb cannot be cultured in the presence of Lyme arthritis joint fluid, this manuscript exclusively analyzes Bb-PG produced by live Bb. The authors do a good job qualifying this correlation by using “likely dead”; however, they might consider experiments that could provide direct evidence. Some ideas include “starving” Bb of nutrients in BSK (ie. growing Bb until cultures become yellow and cells begin to die) or PBS-inactivation of Bb, using this as the input for the pipeline described in Fig 1B.

• Is it possible to characterize Bb-PG directly from the host joint fluid (either murine or human)? While the experiments put forth in this manuscript provide thorough mechanistic data of Bb-PG processing in vitro, it is widely understood that the transcriptional profile (which may affect the varieties of PG produced) is different between in vitro BSK culture and in vivo (most famously expression of OspA vs OspC). Have the authors tried performing LC/MS on joint fluid samples from Lyme arthritis patients (or infected/cured mice)?

• The variability of human samples has been discussed heavily in the manuscript, but a few points stand out:

Bimodal data. For the serum samples in Fig 6B, the green and brown patients versus the red and orange patients cluster in almost a bimodal fashion. However, based on the data presented in Fig 6C, it seems that the distribution of PGLYRP2 ng/mL is evenly distributed. Does this suggest that other enzymes/factors may be at play?

Omission of blue patient. The joint fluid of the blue patient is not displayed in 6B, JF columns in the top row of panels. This datapoint is of particular interest, given the JF PGLYRP2 level relative to the PGLYRP2 level of the red patient in serum (Fig. 6C). If PGLYRP2 is responsible for amidase cleavage, one would expect that the extracted ion counts of the blue JF patient would match the peptide pattern of the red patient in the serum.

Healthy Controls. Measuring PGLYRP2 and Lysozyme (Fig. 6C) levels in JF and serum of healthy controls would be interesting. Could it be possible that the endogenous levels of these enzymes are lower/higher in Lyme patients due to inflammation/infection? One could imagine adding healthy control samples to 6A-C.

**Part III – Minor Issues: Editorial and Data Presentation Modifications**

Reviewer #1: Results:

Lines 157-158. Since the mass spec data were matched against a computational library, it isn't clear how the authors could be certain that the PG fragments had an anhydro modification.

Lines 241-244. Even after multiple readings, this reviewer finds this complex sentence difficult to understand.

Figure 6 and related text in Results. The authors should indicate that their data do not enable them to state definitively that the fragments observed are due to the specific enzymes mentioned.

Discussion:

-Most of the Discussion on p17 seems superfluous.

-Readers would benefit if the current data were explained in a more straightforward fashion in the context of the author's previous PNAS 2019 paper. In other words, which results from the prior study are they building upon in the current work?

-Similarly, the innate and autoimmune mechanisms that presumably collaborate to cause Lyme arthritis could be presented as a model followed by supportive evidence.

Reviewer #2: - Line 207, compared not compares

- Line 255 “subunits”

- Line 238 “increased”

- Line 242 “environment is derived”

- Line 292 “to be poor inducers”

- Line 299 “refer to PG fragments” (also delete “to” later in the same sentence

- Section starting with line 353 – the variability across patients is intriguing. How did you ensure that sampling and sample handling was consistent between all patients to exclude sampling artifacts as a source of variation? The description in the methods does state that samples were treated equally, but I wonder if those samples could be normalized to e.g. total protein content to make sure they were not inadvertently diluted during sampling? Alternative to normalizing, more information on the sampling in the Methods section would help (currently just states that samples were “collected”, but unclear how exactly”

- Line 444 – this is a bit unclear. Since B. burgdorferi can scavenge PG from the environment, it must possess an uptake system and the cytosolic mechanisms to turn PG into a form it can use. How is this different from a canonical recycling pathway, unless the transporter somehow specifically recognizes and transports non-Borrelia PG?

Reviewer #3: • Consistency of naming the control condition in Fig. 1 should be noted: in 1E and throughout the figure 1 legend, the control is referred to as “BSK”, while 1B refers to the control as “Medium only”. Consider changing 1B “BSK control”.

• While the Bb strains used in 2A are all infectious, did the authors perform any plasmid typing confirmation to ensure that these strains possess their full plasmid profile (and therefore carry the entire genome to produce the full spectrum of Bb-PG)?

• The statement made in line 381-383 (The results indicate that while both types of fluid display lysozyme activity...) may be misleading. While 6C does indeed show lysozyme presence (ug/mL, detected by ELISA) in both joint fluid and serum, it seems like activity is absent in serum, given that the extracted ion counts for Bb-PG products produced by lysozyme cleavage (bottom four panels of 6B) show little to no lysozyme cleavage above background levels.

PLOS authors have the option to publish the peer review history of their article (what does this mean? ). If published, this will include your full peer review and any attached files.

**Do you want your identity to be public for this peer review?** For information about this choice, including consent withdrawal, please see our Privacy Policy .

Reviewer #1: No

Reviewer #2: No

Reviewer #3: No

**Figure resubmission:**
---

## [Editor Report · Decision Letter 1]

Christine,

We are pleased to inform you that your manuscript 'Bacterial and host enzymes modulate the pro-inflammatory response elicited by the peptidoglycan of Lyme disease agent Borrelia burgdorferi' has been provisionally accepted for publication in PLOS Pathogens.

Best regards,

John M Leong

Academic Editor

PLOS Pathogens

D. Scott Samuels

Section Editor

PLOS Pathogens

Sumita Bhaduri-McIntosh

Editor-in-Chief

PLOS Pathogens

orcid.org/0000-0003-2946-9497

Michael Malim

Editor-in-Chief

PLOS Pathogens

orcid.org/0000-0002-7699-2064
---

## [Editor Report · Acceptance letter]

Dear Dr Jacobs-Wagner,

We are delighted to inform you that your manuscript, "Bacterial and host enzymes modulate the pro-inflammatory response elicited by the peptidoglycan of Lyme disease agent Borrelia burgdorferi," has been formally accepted for publication in PLOS Pathogens.

Best regards,

Sumita Bhaduri-McIntosh

Editor-in-Chief

PLOS Pathogens

orcid.org/0000-0003-2946-9497

Michael Malim

Editor-in-Chief

PLOS Pathogens

orcid.org/0000-0002-7699-2064